# Latent Execution for Neural Program Synthesis

**Xinyun Chen**
UC Berkeley
xinyun.chen@berkeley.edu

**Dawn Song**
UC Berkeley
dawnsong@cs.berkeley.edu

**Yuandong Tian**
Facebook AI Research
yuandong@fb.com

## Abstract

Program synthesis from input-output (IO) examples has been a long-standing challenge. While recent works demonstrated limited success on domain-specific languages (DSL), it remains highly challenging to apply them to real-world programming languages, such as C. Due to complicated syntax and token variation, there are three major challenges: **(1)** unlike many DSLs, programs in languages like C need to compile first and are not executed via interpreters; **(2)** the program search space grows exponentially when the syntax and semantics of the programming language become more complex; and **(3)** collecting a large-scale dataset of real-world programs is non-trivial. As a first step to address these challenges, we propose LaSynth and show its efficacy in a *restricted-C* domain (i.e., C code with tens of tokens, with sequential, branching, loop and simple arithmetic operations but no library call). More specifically, LaSynth learns the latent representation to approximate the execution of partially generated programs, even if they are incomplete in syntax (addressing **(1)**). The learned execution significantly improves the performance of next token prediction over existing approaches, facilitating search (addressing **(2)**). Finally, once trained with randomly generated ground-truth programs and their IO pairs, LaSynth can synthesize more concise programs that resemble human-written code. Furthermore, retraining our model with these synthesized programs yields better performance with fewer samples for both Karel and C program synthesis, indicating the promise of leveraging the learned program synthesizer to improve the dataset quality for input-output program synthesis (addressing **(3)**). When evaluating on whether the program execution outputs match the IO pairs, LaSynth achieves 55.2% accuracy on generating simple C code with tens of tokens including loops and branches, outperforming existing approaches without executors by around 20%. [1]

## 1 Introduction

Program synthesis from input-output (IO) pairs, also called programming by example (PBE), requires high-level reasoning and remains a challenging problem for deep models. Unlike Natural Language Processing (NLP) [5, 16] and perceptual tasks such as Computer Vision (CV) [14, 22], the mapping from IO pairs to the program itself is hard to model. Many works attempt to learn a direct mapping from training samples, but often found that it is already difficult to achieve a low training error, and generalization to new problems is even harder. Alternatively, one might choose to formulate program synthesis as a search problem: to find the program that satisfies IO pairs. Unfortunately, the search space of programs is often vast and highly non-smooth, i.e., a small perturbation of the program often leads to a complete change of the output.

While there are many previous works on programming by example tasks [6, 17, 9], they mainly focus on Domain Specific Languages (DSLs), and cannot be easily applied to popular general-purpose programming languages. For example, to synthesize C programs, we need to deal with both high-level

---

[1]The code is available at https://github.com/Jungyhuk/latent-execution.

35th Conference on Neural Information Processing Systems (NeurIPS 2021).

control flows (e.g., branching and loop) and low-level operations (e.g., which variable is the target of assignment). Moreover, unlike DSLs (e.g., Karel) for which it is feasible to implement a per-line interpreter, C programs need compilation and a partial C program cannot execute. On the other hand, some recent works investigate natural language descriptions as the auxiliary information of the program specification, and they evaluate neural program synthesis models on constrained or simplified competitive programming problems [27, 3, 23, 11, 4]. Although some of these works demonstrate promising results for synthesizing Python or C code, they require manual annotations of natural language specifications [27] or large-scale pre-training on human-written programs [11, 4], and the performance significantly degrades when only input-output examples are fed into the synthesizer [3].

To synthesize C programs from input-output examples only, we propose `LaSynth`, which generates the program in a recurrent and token-by-token manner. As the first contribution on model architectures for program synthesis, we propose to use two latent *parallel representations* in the recurrent model. One representation is learned from regular recurrent models as in autoregressive language models [24], with the double attention mechanism over IO pairs proposed in RobustFill [17] and an operation predictor that models the arithmetic relationship between the program input and output. The second representation, named *Latent Execution Trace (LaET)*, models the hypothetical input signal for the remaining partial program to execute to get to the desired output. Motivated by the line of work on execution-guided program synthesis [47, 18, 57, 12], we learn a latent representation for C programs which are not executed via interpreters, and train the model given only IO pairs without the intermediate program execution states. The two parallel representations are trained end-to-end.

As the second contribution on dataset construction, we demonstrate that it is possible to automatically construct a C codebase that is of high quality, controllable and concise through our proposed program synthesis procedure. Specifically, starting from randomly generated C programs that might contain a lot of redundant statements, we show that via *iterative retraining*, the subsequent generated code from our learned model becomes more concise and similar to human-written ones. Moreover, learning directly from the generated code leads to better performance given the same amount of samples, and improves the sample efficiency. We observe similar results when applying our iterative retraining technique to Karel [9], another programming by example benchmark consisting of randomly generated programs. Although the initial Karel dataset includes a large proportion of complicated programs with different control flow constructs, we demonstrate that nearly half of the problems can be solved by straight-line programs, which again confirms that randomly generated programs tend to be unnecessarily complicated. We envision that the iterative retraining procedure could greatly reduce laborious efforts in human codebase collection in future research.

As the third contribution, we show for the first time that short C code in a restricted domain (tens of tokens, no library call) with sequential, branching, loop and simple arithmetic operations can be effectively synthesized from IO pairs only. In particular, while `LaSynth` tends to generate more concise programs (and does not have exact token match with random generated ground truth code), when measuring whether the program execution outputs match the IO pairs, `LaSynth` achieves $55.2\%$ accuracy, and outperforms existing neural program synthesis models by around $20\%$. These results demonstrate the effectiveness of learning latent execution traces.

## 2 Neural Program Synthesis from Input-Output Examples

In programming by example tasks, the program specification is a set of input-output examples [17, 9]. Specifically, we provide the synthesizer with a set of $K$ input-output pairs $\{(I^{(k)}, O^{(k)})\}_{k=1}^K$ ($\{IO\}^K$ in short). These input-output pairs are annotated with a ground truth program $P^\star$, so that $P^\star(I^{(k)}) = O^{(k)}$ for any $k \in \{1, 2, ..., K\}$. To measure the program correctness, we include another set of held-out test cases $\{IO\}_{test}^{K_{test}}$ that differs from $\{IO\}^K$. The goal of the program synthesizer is to predict a program $P$ from $\{IO\}^K$, so that $P(I) = P^\star(I) = O$ for any $(I, O) \in \{IO\}^K + \{IO\}_{test}^{K_{test}}$.

**C Program Synthesis**. In this work, we make the first attempt of synthesizing C code in a restricted domain from input-output examples only, and we focus on programs for list processing. List processing tasks have been studied in some prior works on input-output program synthesis, but they synthesize programs in restricted domain-specific languages instead of full-fledged popular programming languages [6, 39, 38].

Our C code synthesis problem brings new challenges for programming by example. Compared to domain-specific languages, the syntax and semantics of C are much more complicated, which significantly enlarges the program search space. Meanwhile, learning good representations for partially

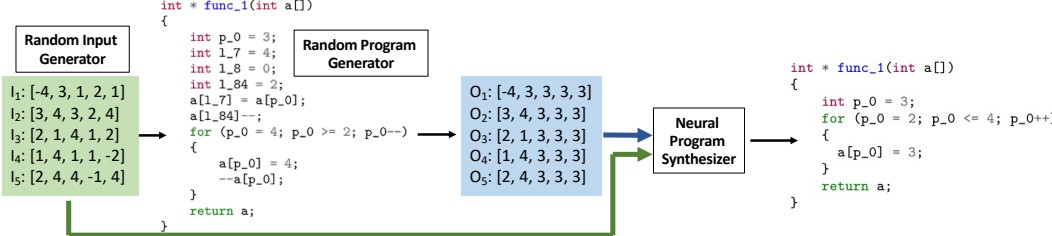

Figure 1: Illustration of the C program synthesis pipeline. For dataset construction, we develop a random program generator to sample random C programs, then execute the program over randomly generated inputs and obtain the outputs. The input-output pairs are fed into the neural program synthesizer to predict the programs. Note that the synthesized program can be more concise than the original random program.

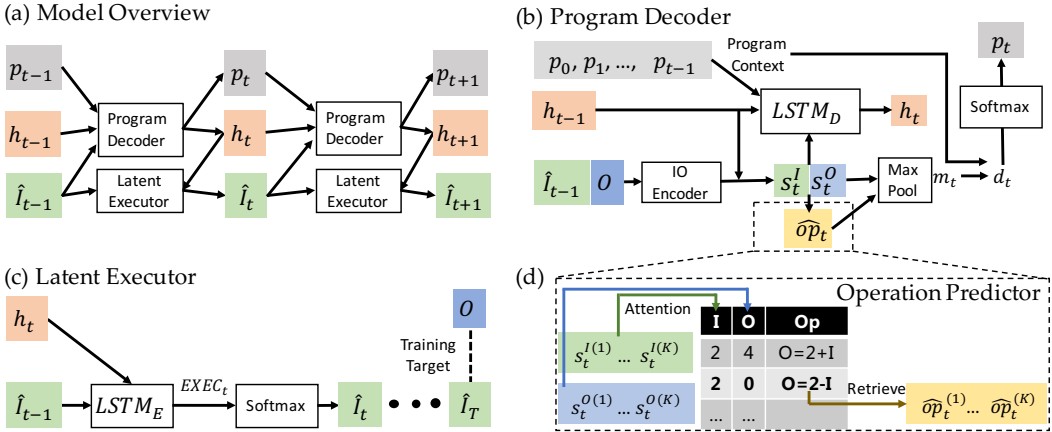

Figure 2: (a) An overview of `LaSynth` model architecture. (b), (c), and (d) present the details of the program decoder, latent executor, and the operation predictor. Note that the operation predictor is specialized for numerical calculation, and thus is not used for the Karel domain.

decoded programs also becomes more difficult. In particular, prior neural program synthesizers that utilize per-line interpreters for the programming language to guide the synthesis and representation learning [12, 44, 37, 18, 38] are not directly applicable to C. Although it is possible to dump some intermediate variable states during C code execution [10], since partial C programs are not executable, we are able to obtain all the execution states only until a full C code is generated, which is too late to include them in the program decoding process. In particular, the intermediate execution state is not available when the partial program is syntactically invalid, and this happens more frequently for C due to its syntax design.

## 3 Program Synthesis with Learned Execution

In this section, we present `LaSynth` which learns to represent the execution of partial programs to guide the synthesis process. Fig. 2(a) provides an overview of `LaSynth` model architecture which consists of two components, the *program decoder* and the *latent executor*. We present the core design below, and defer more details to Appendix B and Appendix C.

### 3.1 Model Overview

At a high level, the program decoder (Fig. 2(b)) takes a latent vector $h_{t-1}$ that represents the generated partial program, the previous (generated) program token $p_{t-1}$, and outputs the latent vector $h_t$ and the next program token $p_t$ to be generated at time step $t$:

$$(h_t, p_t) = \text{ProgramDecoder}(h_{t-1}, p_{t-1}; IO_{t-1}) \tag{1}$$

Here the recurrent model is conditioned on the IO pair $IO_{t-1}$. When $IO_t = IO := (I, O)$ for every $t$, i.e., $IO_t$ remains *constant* over the entire recurrent generation process, Eqn. 1 represents the standard recurrent architecture used in most autoregressive natural language models [24, 49], and is also used in prior works on program synthesis from input-output examples [17, 9].

For program decoding, the decoder first takes two attention vectors $s_t^I$ and $s_t^O$ computed from IO pairs and latent vector $h_{t-1}$ via double attention [17], and utilizes a max pooling layer to compute an aggregated embedding $m_t$ for all IO pairs (Fig. 2(b)):

$$m_t = \text{MaxPool}_{j \in \{1,2,...,K\}}(\tanh(W[s_t^{I(j)}; s_t^{O(j)}])) \tag{2}$$

Here the superscript $(j)$ indicates that the representation is for the $j$-th IO pair, $[a; b]$ is vector concatenation of $a$ and $b$, and $W$ is a trainable matrix. To facilitate the prediction of long programs, we compute an attention vector $d_t$ over previously generated program tokens using the standard attention mechanism [5, 30]:

$$d_t = \text{Attention}(m_t, \{p_0, ..., p_{t-1}\}) \tag{3}$$

Finally, the next token $p_t$ is sampled from $\mathbb{P}[p_t] = \text{Softmax}(V d_t)_{p_t}$ where $V$ is a trainable matrix.

### 3.2 Latent Executor Design

As shown in our experiments (Sec. 5), the standard program decoder architecture may not be able to achieve strong performance in program synthesis when the program complexity increases. One main reason is that the standard program decoder only takes the initial IO pairs as the input without considering the program execution, thus the learned representation for the partial program does not effectively guide the synthesis process. Motivated by prior works that utilize execution traces for Karel program synthesis [12, 44, 47], in this paper, we introduce *latent executor* (Fig. 2(c)) which maintains a second representation $\hat{I}_t$ during program decoding. Intuitively, $\hat{I}_{t-1}$ models the *hypothetical input* of the partial program $p_{t...T}$ so that its output becomes $O$. Given the estimated input $\hat{I}_{t-1}$ and the latent vector $h_t$, the latent executor returns $\hat{I}_t$ at the next time step $t$:

$$\hat{I}_t = \text{LatentExecutor}(\hat{I}_{t-1}, h_t) \tag{4}$$

The collection of $\{\hat{I}_t\}_{t=0}^T$ is the *latent execution trace (LaET)*. With the help of latent executor, we now use the IO pairs $IO_{t-1} := (\hat{I}_{t-1}, O)$ instead of $(I, O)$ for the program decoder (Eqn. 1).

### 3.3 End-to-end Training

We train our model with supervised learning, by minimizing the sum of token prediction loss $\mathcal{L}_{Prog}$, and the latent executor loss $\mathcal{L}_{Exec}$:

$$\mathcal{L} = \mathcal{L}_{Prog} + \mathcal{L}_{Exec} \tag{5}$$

Specifically, $\mathcal{L}_{Prog} := \sum_{t=1}^{T} \text{Loss}(p_t, p_t^{\star})$ is the step-by-step cross-entropy loss between the predicted programs $p_{1...T}$ and the ground truth programs $p_{1...T}^{\star}$.

For latent executor, since the semantics of partial programs (e.g., partial C programs) are not always well-defined, there is no step-by-step training supervision. However, the output of the executor should be consistent with the program specification after taking the annotated ground truth program as the input. Therefore, we set $\hat{I}_0 = I$ (true input) and minimize the distance between $\hat{I}_T$ and $O$ (true output) after the program finishes:

$$\mathcal{L}_{Exec} = \text{Loss}(\hat{I}_T, O) \tag{6}$$

Note that $\mathcal{L}_{Exec}$ does not rely on any assumptions of the partial program semantics, and thus is applicable to both domain-specific languages and general-purpose programming languages such as C. In our evaluation, equipping with the latent executor significantly improves the program prediction performance, where each program could include up to 256 tokens.

### 3.4 Data Regeneration and Iterative Retraining

Interestingly, once our model is trained on the initial random generated programs $\mathcal{D}_0$, the predicted program becomes more concise and resembles human-written code. While the exact token match accuracy is low even on the training set, the model still satisfies the IO pairs for many problems. We leverage such a phenomenon to construct a new dataset $\mathcal{D}_1$ with higher-quality programs from $\mathcal{D}_0$. Specifically, we run beam search on the trained model to predict program $p_{0...T}$ given input-output pairs in the training set. If model prediction $p_{0...T}$ satisfies all the input-output examples and held-out cases, we replace the original program $p_{0...T}^{\star}$ with $p_{0...T}$ in $\mathcal{D}_1$, and keep $p_{0...T}^{\star}$ otherwise. Afterward, we re-train the model on $\mathcal{D}_1$. In Sec. 5, we will demonstrate that the retraining process further improves the model performance, especially with smaller training datasets.

Table 1: The comparison between our restricted C domain and existing programming by example tasks.

| | Control flow | Variables | Arithmetics | No helper functions |
|---|:---:|:---:|:---:|:---:|
| Restricted C (Ours) | ✓ | ✓ | ✓ | ✓ |
| Karel [9] | ✓ | – | – | – |
| DeepCoder [6] | – | ✓ | ✓ | – |
| FlashFill [19] | – | – | – | – |

## 4 Restricted C Program Synthesis Domain

In this section, we discuss our restricted C program synthesis domain, and our operation predictor design for improving the numerical reasoning ability of program synthesis models.

### 4.1 Data Generation

Collecting large-scale high-quality datasets for program synthesis requires a lot of human efforts, and we aim to reduce the manual work for dataset construction.

Our data generator is built upon Csmith [54], a random C code generation tool originally designed for finding bugs in compilers. Following the common practice of generating input-output pairs, for each program, we randomly sample 5 numerical lists as the program inputs, and execute the program to obtain the corresponding output lists. This is similar to existing works on PBE problems that sample programs based on a probabilistic context-free grammar, randomly generate valid inputs for the programs and obtain the outputs [40, 15, 6]. This creates infinite samples for synthesizing programs in domain-specific languages. While the programs sampled in this way differ from human-written code, Sec. 3.4 shows that they can be converted to be more concise and human-like.

**The subset of language features used**. Our generated program has variable declaration, variable assignment, and expressions with addition or subtraction operations. The programs also have non-sequential statements, including If statements, For loops, Continue and Break statements. Except for the input argument which is a list, all variables declared are integers, and all program statements are integer manipulation. Each expression has at most 2 mathematical operations, and chaining the full C program could perform multi-step numerical calculation (e.g., p0 = p0 - p1 + p2; p0 = p0 - 1;). Looping statements other than For (i.e., While or Do-While loops) are not supported. Note that we only constrain the final program length ($\leq 256$ tokens) and the program can have nested for-loops and complicated if-conditions.

**Post-processing**. We perform a few post-processing steps to obtain our final programs from programs generated by Csmith (see Fig. 1 for an example). We resample different components of the program, so that (1) each constant numerical value lies in $[-4, 4]$, (2) mathematical operators only contain addition and subtraction, and (3) upper/lower limits of For loops are positive and within the length of the list. Programs are discarded if they are trivial (e.g., constant or identity mappings), or the input-output examples include values out of the range $[-4, 4]$.

**Final dataset**. We reweight the program distribution so that at least half of them include For loops. Our full dataset includes $500K$ samples in the training set, $1K$ samples in the validation set, and $1K$ samples in the test set. As shown in Fig. 1, the randomly sampled program may contain redundant statements, which can be easily avoided by human programmers. We compare our restricted C domain to prior datasets of programming by example in Table 1.

### 4.2 Program Decoding with the Operation Predictor

For program decoder, predicting the next program token $p_t$ is non-trivial, especially when mathematical reasoning is required [43, 29]. To improve the program synthesis performance for domains involving numerical calculation, such as our restricted C domain, we design an associative memory structure named *operation predictor* (Fig. 2(d)), based on the following intuition: given the input $I = 2$ and output $O = 4$, human would infer that "$O = I + 2$" might be the desired operation and write down the code accordingly. To materialize such an intuition, we create a pre-computed table that covers all possible integer addition and subtraction operations for valid input and output list values. We defer the details of the model architecture to Appendix B.2. The program decoding process remains similar to the one described in Sec. 3, and we highlight the key differences as follows.

Table 2: The comparison between `LaSynth` and baseline neural program synthesis models in our evaluation.

| | LaSynth | Exec [12] | Shin et al. [44] | Bunel et al. [9] | RobustFill [17] | Property Signatures [39] |
|---|---|---|---|---|---|---|
| + Program execution | ✓ | ✓ | ✓ | – | – | – |
| No interpreter needed | ✓ | – | – | ✓ | ✓ | ✓ |

The operation predictor takes two attention vectors $s_t^I$ and $s_t^O$ as the representations of input-output examples, and yields an operator embedding $\hat{op}_t$. To compute the aggregated embedding vector for all input-output examples, we modify Eqn. 2 to also take $\hat{op}_t$ as an input of the max pooling layer:

$$m_t = \text{MaxPool}_{j \in \{1,2,...,K\}}(\tanh(W[s_t^{I(j)}; s_t^{O(j)}; \hat{op}_t^{(j)}])) \tag{7}$$

To train the operation predictor, we add an additional loss $\mathcal{L}_{Op}$:

$$\mathcal{L} = \mathcal{L}_{Prog} + \mathcal{L}_{Exec} + \mathcal{L}_{Op} \tag{8}$$

$\mathcal{L}_{Op}$ is designed to ensure that the operation predictor predicts operations related to IO pairs, and we defer the details to Appendix B.2.

**Limitations.** In our current implementation of the operation predictor, the operation table is only able to enumerate the arithmetic operations over a pre-defined constant set, thus it requires that the set of possible numerical values in input-output pairs is finite. One way of extending our operation predictor to support potentially unbounded numerical calculation is to combine it with the subword tokenizer, which has been commonly used in recent language models [16, 11, 4]. We consider designing general-purpose number representation for better mathematical reasoning as future work.

## 5 Experiments

In this section, we discuss our results on synthesizing programs in Karel and C languages. We first show that `LaSynth` achieves competitive performance on Karel benchmark. Then we present the results on our restricted C benchmark, and demonstrate that our approach significantly outperforms existing neural program synthesis models. Finally, we discuss the effect of iterative retraining.

### 5.1 Karel Program Synthesis

#### 5.1.1 Evaluation Setup

**Karel domain.** Karel is an educational programming language [41], and has been studied in recent works on neural program synthesis from input-output examples [15, 9, 12, 44]. A Karel program controls a robot in a 2D grid world. There are instructions that control the robot, e.g., move, `turnLeft` and `PutMarker`, as well as conditionals and loops, i.e., `if`, `repeat` and `while`. See Appendix A for grammar specification and the state representation.

We train and evaluate all models on the Karel dataset introduced in [9]. The dataset contains randomly sampled programs from the Karel DSL ($1.1M$ training samples, $2.5K$ samples in the validation set and $2.5K$ samples in the test set). Each program includes 5 input-output pairs as the specification, and the sixth pair as the held-out test case. Following the prior work, we evaluate two metrics: (1) **Exact Match**: the predicted program is the same as the ground truth; (2) **Generalization**: the predicted program satisfies both the input-output pairs and the held-out input-output test case.

**Baselines.** *Bunel et al.* [9] designed the first program synthesis model for the Karel benchmark with a similar high-level design as RobustFill, but they use convolutional neural networks (CNN) to encode the Karel grid maps. Compared to `LaSynth`, this model does not utilize any program execution information, and does not include our latent executor. Instead of directly synthesizing the program from input-output examples, the model in *Shin et al.* [44] first predicts the execution traces containing the robot actions from the input-output pairs, then decodes the program based on the execution traces. This model improves the prediction performance over Bunel et al., but it requires the full execution traces for model training and an interpreter for execution. *Exec* [12] leverages the execution states of partial generated programs to guide the subsequent synthesis process, but the execution states are obtained from the Karel interpreter rather than learned by the model, thus this approach represents the ideal scenario where the partial programs could be executable.

Table 3: Results on Karel dataset. **Gen** and **Exact** denote generalization and exact match accuracies.

| Approach | Gen | Exact |
|---|---|---|
| LaSynth | 83.68% | 41.12% |
| Exec [12] | **86.04%** | 39.40% |
| Bunel et al. [9] | 77.12% | 32.17% |
| Shin et al. [44] | 81.30% | **42.80%** |

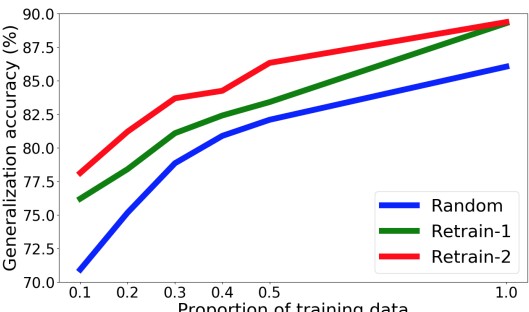

Figure 3: Generalization accuracies with different training data sizes on Karel. With the full training set, the accuracies are $86.04\%$, $89.28\%$ and $89.36\%$ for training on random programs, retraining for 1 and 2 iterations.

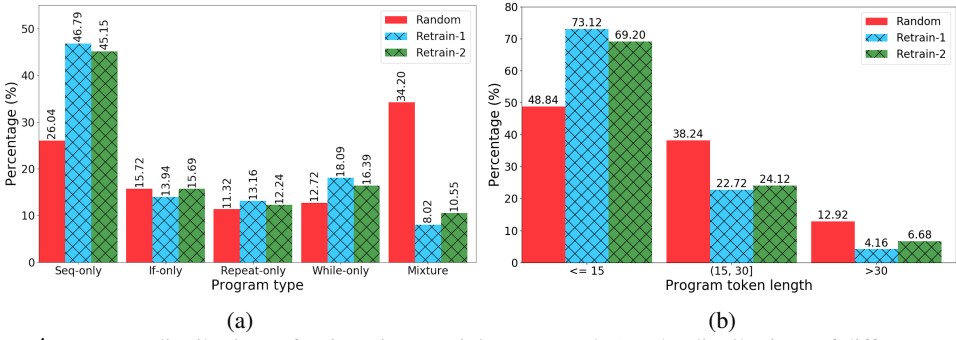

(a)          (b)

Figure 4: Program distributions after iterative retraining on Karel. (a) The distributions of different program types. *Seq-only*: no control flows. *If-only*: the program includes If statements but no loops. *Repeat/While-only*: the program includes Repeat/While loops, but no other control flow constructs. *Mixture*: the program includes at least two types of control flow constructs. (b) The distributions of programs with different token lengths.

Our model architecture for Karel is largely similar to the model for C code synthesis, except that we employ the CNN encoder in Bunel et al. [9] in our program decoder and latent executor. The comparison with baseline models is shown in the middle block of Table 2. All models use the beam search for decoding programs, with the beam size of 64.

### 5.1.2 Results

We present the results of `LaSynth` and baseline model architectures in Table 3. First, `LaSynth` outperforms all baselines that do not incorporate the partial program execution information, and achieves competitive performance compared with the Exec algorithm that requires an interpreter to obtain the partial program execution states. In particular, `LaSynth` achieves a higher generalization accuracy than Shin et al. with lower exact match accuracy, showing that decoded programs by `LaSynth` are more different from randomly generated programs. Although Shin et al. also model the program execution by predicting the robot actions, the prediction of the action traces does not take the program structure into account, resulting in the inferior performance.

## 5.2 C Code Synthesis

### 5.2.1 Evaluation Setup

Given the variety of C programs, we observe that the exact match accuracies of models are mostly nearly 0. Therefore, we focus on evaluating the generalization accuracy, and we consider the predicted program to be correct when it satisfies both the 5 input-output examples and 5 held-out test cases.

**Baselines.** We compare the full `LaSynth` with its multiple ablated versions:

- `NoExecutor`. The program decoder (Eqn. 1) always takes the initial input-output pairs as the input; i.e,. $\hat{I}_t = I_0$ for every $t$.

```
int * func_1(int a[])
{
    int p_0 = 0;
    int l_25 = 4;
    a[p_0] = 1;
    --a[l_25];
    return a;
}
```

```
int * func_1(int a[])
{
    int p_0 = 2;
    int l_12 = 3;
    for (p_0 = 1; p_0 <= 2; p_0++)
    {
        a[p_0]--;
    }
    a[l_12] = a[l_12] + 4;
    return a;
}
```

```
int * func_1(int a[])
{
    int p_0 = 0;
    int l_7 = 3;
    int l_8 = 1;
    a[l_8] = (a[l_7] - a[p_0]);
    for (p_0 = 3; p_0 <= 4; p_0++)
    {
        for (int p_1 = 1; p_1 <= 2; p_1++)
        {
            a[p_1] = a[p_1] + a[p_0];
            a[p_1] = a[p_1] + 2;
        }
    }
    return a;
}
```

Figure 5: Sample programs that could be correctly predicted by `LaSynth`, but wrongly predicted by models without the latent executor. These programs require multiple different operations for different input list elements.

- `NoPartialExecutor`. $\hat{I}_t = I_0 = I$ for every $t$ and additionally $h_T$ is regularized so that $\text{LatentExecutor}(I_0, h_T)$ matches the output $O$ under loss $\mathcal{L}_{Exec}$. Therefore, no partial latent execution.
- `NoOpPredictor`. The max pooling layer only takes the vectors computed by the double attention as the input (Eqn. 2).
- `NoAttentionInDecoding`. There is no attention over decoded program tokens, and the output of the max pooling layer is directly fed into the output softmax layer; i.e., $\mathbb{P}[p_t] = \text{Softmax}(V m_t)_{p_t}$ (compared to Eqn. 3).

We also compare with existing neural program synthesis models with good performance on related tasks, as shown in the rightmost block of Table 2. *RobustFill* [17] is the state-of-the-art neural network architecture on FlashFill benchmark, which synthesizes string manipulation programs in a domain-specific language. As described in Sec. 3, the input-output encoder and the program decoder architectures in RobustFill are similar to `LaSynth`, except that it does not include the latent executor, operation predictor, and the attention on the decoded program sequence.

*Property Signatures* [39] was designed for synthesizing list manipulation programs in domain-specific languages, but instead of taking the raw input and output lists as the neural network input, they design some properties that distinguish different programs, then take the values of these properties as the model input. A sample property could be whether the program output is the same as the input, and the property values could be "All True", "All False", or "Mixed", indicating that the property always holds for any input-output pair in the specification, never holds, and holds for some pairs but not others, respectively. We customize the original design [39] for our setting. First, our property set takes the format of $O = C + I$? and $O = C - I$?, where $C \in [-4, 4]$. For example, $O = 2 + I$? means whether the output $O$ could be calculated by adding 2 to the input $I$. These properties focus more on numerical calculation, similar to our operation predictor. Second, different from the task in [39], our C programs sometimes manipulate only a subset of the input lists, thus encoding the list with a single property value is inappropriate. Instead, we compute the property value per element in input-output pairs, use a bi-directional LSTM to encode the property values as a sequence, then take the outputs of the bi-LSTM for program prediction.

### 5.2.2 Results

Table 4 presents the results, where all models are trained on the initial random programs. The full `LaSynth` outperforms other variants, and improves the performance of RobustFill and Property Signatures by around 20%. We also increase the model size of RobustFill to see if the improvement comes from larger model size, but the results are not better. In particular, the latent executor significantly increases the prediction accuracy, and achieves better results than `NoPartialExecutor`, which shows that learning latent execution traces leads to better partial program representations. In Fig. 5, we present sample programs that could be correctly synthesized by `LaSynth`, but models without the latent executor provide the wrong prediction. We observe that the latent executor is beneficial when the program involves different manipulations for different list elements, e.g., more than one For loop and different mathematical calculations. Our breakdown results on programs of different complexity also justify this observation. We first present the results on programs with different control flow constructs in Fig. 6. Specifically, *Seq-only* includes programs with no control

Table 4: Results on C dataset.

| Approach | Accuracy |
|---|---|
| LaSynth | **55.2%** |
| NoAttentionInDecoding | 53.5% |
| NoOpPredictor | 53.7% |
| NoPartialExecutor | 42.9% |
| NoExecutor | 38.6% |
| RobustFill [17] | 37.6% |
| Property Signatures [39] | 34.5% |

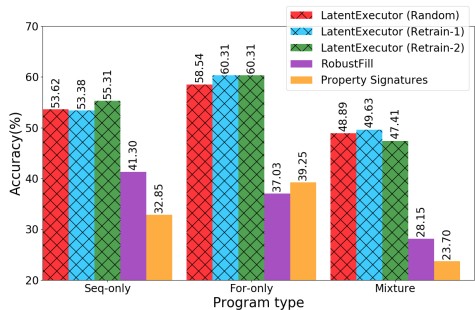

Figure 6: Accuracies of different program types on C dataset.

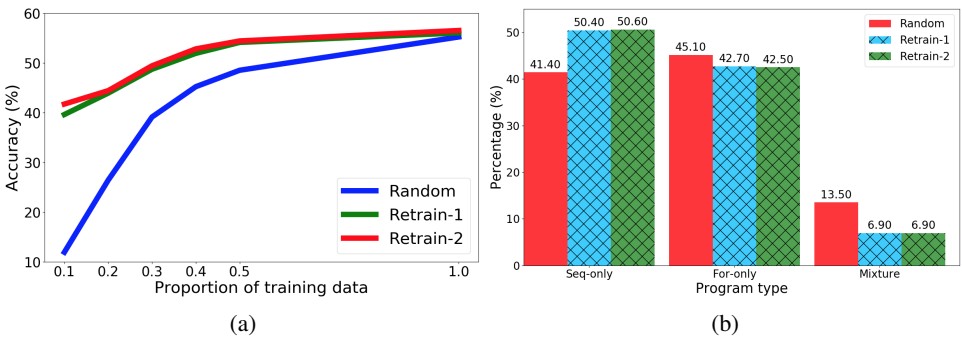

(a)                                         (b)

Figure 7: Results of iterative retraining on the C dataset. (a) Accuracies with different training data sizes. With the full training set, the accuracies are 55.2%, 56.0% and 56.5% for training on random programs, retraining for 1 and 2 iterations, respectively. (b) The program distributions after each retraining iteration.

flow constructs, *For-only* includes programs with For loops but no If statements, and *Mixture* includes programs with both For loops and If statements. Then we demonstrate the results on different program lengths in Fig. 8b. We show that LaSynth achieves decent performance on long and complicated programs, while the accuracies of baseline models drop dramatically.

### 5.3 Discussion of Iterative Retraining

In Fig. 3, we show the effectiveness of retraining on decoded Karel programs (Sec. 3.4). We observe that retraining for one iteration is sufficient, and it significantly improves the generalization accuracy by over 3%. To understand the differences between predicted programs and randomly generated programs, we demonstrate the changes of dataset distributions after each retraining iteration in Fig. 4a and 4b. We observe that the model learns to predict more concise programs than the ground truth for a large proportion of input-output examples, and considerably alters the dataset distribution so that it becomes more concentrated on short programs with simplified control flow structures. Specifically, from Fig. 4a, although the initial Karel dataset seems to include a large proportion of complicated programs with different control flow constructs, our model synthesizes straight-line programs for nearly half of the samples, which means that many loops and branches in the annotated ground truth programs are unnecessary. This distribution shift also explains the gap between the exact match and generalization accuracies. The program distribution after the second retraining iteration is largely similar to the first iteration, thus retraining for more iterations does not considerably improve the performance. Note that in the second iteration, the synthesizer tends to generate slightly more complicated programs than the first iteration, in order to deal with the cases when the input-output examples oversimplify the intended program functionality. For example, sometimes the input-output examples do not cover the edge cases that the robot may encounter, thus adding additional If branches could avoid the crashes when testing on held-out cases.

Fig. 7a presents the results of retraining on decoded C programs. Similarly, retraining improves the prediction accuracy, especially when the training set is small. From Fig. 7b and 8a, we again observe that the model tends to predict shorter programs than the random code, and it eliminates unnecessary control flows to simplify the programs. We present more examples in Appendix D.

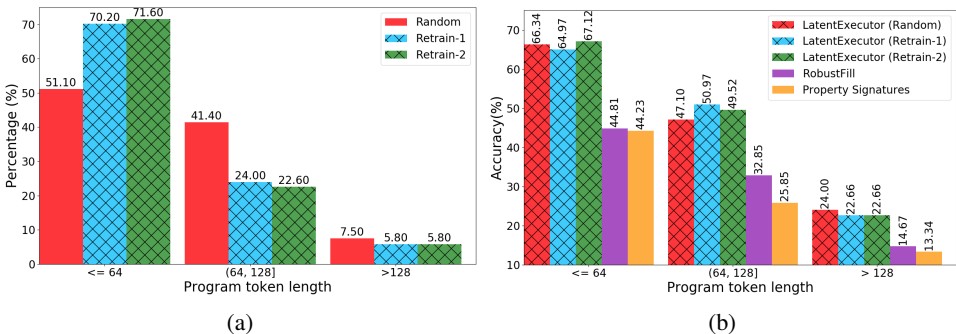

Figure 8: Results on programs of different token lengths on the C dataset. (a) The program token length distributions after each retraining iteration. (b) The accuracies on programs of different token lengths.

# 6 Related Work

**Programming by example.** Programming by example problems have been widely studied with various applications, and recent works have developed deep neural networks as program synthesizers [20, 40, 17, 9]. Most prior works focus on synthesizing programs in domain-specific languages, such as FlashFill [40, 17, 51] for string transformation, Karel [9, 44, 12, 21] for simulated robot navigation, and LISP-style languages for list manipulation [6, 25, 57, 36]. In this work, we make the first attempt of synthesizing C code in a restricted domain from input-output examples only, and we focus on the list manipulation domain.

Some recent works investigate the limitations of synthetic datasets and the ambiguity in program specifications for neural program synthesis [45, 13, 46, 28]. These works focus on reducing the bias of data distributions and generating more diverse input-output pairs, while our data regeneration aims to improve the quality of programs. We consider incorporating both lines of work to further improve the dataset quality as future work. In addition, drawing the inspiration from self-training and bootstrapping techniques developed for other applications [35, 1, 32, 52] to extend our iterative retraining scheme is also another future direction.

**Execution-guided program synthesis.** To learn better program representations, some recent works incorporate the execution information to guide the synthesis process [47, 57, 44, 12, 18, 48, 7, 21, 38, 37, 31]. In particular, leveraging partial program execution states improves the performance for several program synthesis tasks [12, 57, 18, 37]. However, existing approaches rely on program interpreters to provide the intermediate execution results whenever applicable. In contrast, we demonstrate that our latent executor learns the latent execution traces (LaET) without such a requirement. Besides program synthesis, execution traces have also been utilized for other software engineering applications [2, 33].

**Neural execution.** Our latent executor is related to prior works on learning to execute algorithms [55, 50, 53] and programs [8]. They focus on predicting execution results for full algorithms and programs, but do not utilize them for program synthesis. Latent state prediction has also been studied in other applications such as task-oriented dialogue systems [34, 56] and robotics [42].

# 7 Conclusion

In this work, we propose `LaSynth`, which learns the latent representation to approximate the execution of partial programs, even if their semantics are not well-defined. We demonstrate the possibility of synthesizing elementary C code from input-output examples only, and leveraging learned execution significantly improves the prediction performance by around $20\%$. Meanwhile, compared to the randomly generated programs, `LaSynth` synthesizes more concise programs that resemble human-written code, and training on these synthesized programs further improves the prediction performance for both Karel and C program synthesis. Our results indicate the promise of leveraging the learned program synthesizer to improve the dataset quality for programming by example tasks.

We consider extending our approach to synthesize more complicated real-world code as future work. For example, we will integrate our latent executor into large-scale pre-trained language models, which could further improve the performance of those program synthesis models taking natural language specifications. We will also study program synthesis problems with unbounded input ranges and different type signatures, which could be approached with the usage of subword tokenizers.

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
