## A  More Descriptions of the Karel Domain

We present the full grammar of the Karel language in Fig. 9.

To represent the execution states, each Karel grid world has a maximum size of $18 \times 18$, and each cell in the grid world is represented by a 16-dimensional vector corresponding to the features in Table 5. Therefore, each grid world is represented as a $16 \times 18 \times 18$ tensor.

```
Prog p   ::=   def run() : s
Stmt s   ::=   while(b) : s | repeat(r) : s |  s₁ ; s₂ | a
         |     if(b) : s | ifelse(b) : s₁ else : s₂
Cond b   ::=   frontIsClear() | leftIsClear() | rightIsClear
         |     markersPresent() | noMarkersPresent() | not b
Action a ::=   move() | turnRight() | turnLeft()
         |     pickMarker() | putMarker()
Cste r   ::=   0 | 1 | ... | 19
```

Figure 9: Grammar for the Karel task.

| |
|---|
| Robot facing North |
| Robot facing East |
| Robot facing South |
| Robot facing West |
| Obstacle |
| Grid boundary |
| 1 marker |
| 2 markers |
| 3 markers |
| 4 markers |
| 5 markers |
| 6 markers |
| 7 markers |
| 8 markers |
| 9 markers |
| 10 markers |

Table 5: Representation of each cell in the Karel state.

## B  Details in Model Architecture

### B.1  Program Decoder

Our model follows the encoder-decoder framework in prior work on neural program synthesis from input-output examples [17, 9], which includes an encoder for the input-output pairs, and a decoder to synthesize the program.

The program decoder is an LSTM (denoted as $\mathrm{LSTM}_D$), which decodes the program as a token sequence. Let $p_{t-1}$ be the decoded program token at step $t - 1$, $E_p(p_{t-1})$ be the embedding vector of $p_{t-1}$, $h_{t-1}$ be the hidden state of the program decoder at step $t - 1$, and $\hat{I}_{t-1}$ and $O$ be the sequences of vectors representing the input list elements and output list elements. We first compute attention vectors over both the input and output lists, following the double attention mechanism in RobustFill:

$$s_t^O = \mathrm{Attention}(h_{t-1}, O), \quad s_t^I = \mathrm{Attention}([h_{t-1}; s_t^O], \hat{I}_{t-1})$$

The notation $[a; b]$ means the concatenation of vectors $a$ and $b$. Then we calculate the output vector of the program decoder at step $t$ as $h_t = \mathrm{LSTM}_D(h_{t-1}, [E_p(p_{t-1}); s_t^I; s_t^O])$.

**Input-Output Encoder.** For C program synthesis, our input-output encoder architecture is similar to RobustFill [17]. For each input-output pair, we use two bi-directional LSTMs [24] to encode the input and output lists respectively. To capture the relationship between the input and output lists, the output list encoder computes attention vectors over the input list elements, using the standard attention mechanism [5, 30]. We also employ different encoder architectures for program synthesis tasks with other formats of input-output examples, as discussed in Sec. 5.

To capture the required arithmetic operation to convert from the program input to the output, we also include the output of the operation predictor $\hat{op}_t$ for program decoding, and we discuss the details later. Afterwards, the max pooling layer aggregates the representation of different IO pairs to generate a single vector:

$$m_t = \text{MaxPool}_{j \in \{1,2,...,K\}}(\tanh(W[s_t^{I(j)}; s_t^{O(j)}; \hat{op}_t^{(j)}]))$$

Here the superscript $(j)$ indicates that the representation is for the $j$-th IO pair, and $W$ is a trainable weight matrix.

To facilitate the prediction of long programs, we compute an attention vector over previously generated program tokens as follows:

$$d_t = \text{Attention}(m_t, \{E_p(p_0), E_p(p_1), ..., E_p(p_{t-1})\})$$

Finally, the next token $p_t$ is sampled from $\mathbb{P}[p_t] = \text{Softmax}(Vd_t)_{p_t}$ where $V$ is a trainable matrix.

## B.2 Operation Predictor for Restricted C Domain

Training neural networks for mathematical reasoning is a challenging problem itself [43, 29], and jointly predicting the mathematical calculations and other program operations imposes extra burden on the program decoder. To mitigate the difficulty, we include a pre-computed table as part of the model input, which describes possible mathematical operations to derive an output value given the input number. For example, Fig. 2(d) shows that by applying the $O = 2 + I$ operation to the input $I = 2$, the output $O = 4$. For each valid input list value $C$, we include two operations $O = C + I$ and $O = C - I$ in the table. Then for each operation $O = C + I$, we enumerate all valid integer list values $I$, and we include the row $[O = C + I, I, O]$ in the table when $O$ is also within our bounded range. In this way, the table covers all possible integer addition and subtraction operations for valid input and output list values.

With the pre-computed table, the operation predictor aims to predict the most possible program operation at the next step. First, we re-use the same embedding matrices as those in the input-output encoder, and compute the embedding vectors for each numerical element in the table. Let $R$ be the number of table rows. For the $i$-th row, we refer to the embedding vector of the input and output values as $r^{[i]}$ and $c^{[i]}$, respectively. Then we utilize $s_t^I$ and $s_t^O$ to compute the attention weights over the table columns of input and output values as follows:

$$wi_t^{[i]} = \text{AttentionWeight}(s_t^I, \{r^{[i]} | i \in \{1, 2, ..., R\}\})$$

$$wo_t^{[i]} = \text{AttentionWeight}(s_t^O, \{c^{[i]} | i \in \{1, 2, ..., R\}\})$$

Let $op^{[i]}$ be the operation in row $i$, then the probability of selecting the operation in the $i$-th row at step $t$ is

$$\mathbb{P}[op_t = op^{[i]}] \propto wi_t^{[i]} \cdot wo_t^{[i]}$$

Let $E_{op}(op)$ be the embedding vector of the operation $op$, then the operation predictor output is

$$\hat{op}_t = \sum_i \mathbb{P}[op_t = op^{[i]}]E_{op}(op^{[i]})$$

To train the operation predictor, we provide the training supervision at step 0, when no transformation has been applied to the program input:

$$\mathcal{L}_{Op} = \text{Loss}(wi_0^{[i]}, \mathbb{1}[r^{[i]} = \hat{I}_0 = I]) + \text{Loss}(wo_0^{[i]}, \mathbb{1}[c^{[i]} = O]) \tag{9}$$

## B.3 Latent Executor

In RobustFill, the encoder only takes the initial input-output pairs as the input. On the other hand, in recent work on execution-guided program synthesis [12, 47, 57, 18, 38, 37], the execution states of partial programs are leveraged as the model input to guide the subsequent program prediction. However, existing approaches mostly assume that the programs are sequential [57, 18], or require an interpreter of partial programs [12]. To address these limitations, Nye et al. design neural networks to represent the partial program semantics when they are not well-defined [37]. However, they need to train a separate neural module to represent each program operation, thus it is hard to scale beyond domain-specific languages.

In this work, we include another LSTM to approximate the program execution states, denoted as $\text{LSTM}_E$. Let $\hat{I}_{t-1}$ be the input of $\text{LSTM}_E$, which is the program input at step $t - 1$. The output of $\text{LSTM}_E$ is:

$$\text{Exec}_t = \text{LSTM}_E(h_t, \hat{I}_{t-1})$$

### B.3.1 Implementation for Restricted C Domain

For our restricted C domain, the length of $\text{Exec}_t$ is the same as $\hat{I}_{t-1}$, i.e., the input list length. Let $L$ be the length of input and output lists. Let $\mathbb{P}[I_t = v]$ be the probability that the execution result at step $t$ is $v$, then:

$$\mathbb{P}[I_{t,l} = v_l] = \text{Softmax}(W_E \text{Exec}_{t,l})_{v_l}$$

Here the subscript $l$ denotes that the representation is for the $l$-th list element, and $W_E$ is a trainable weight matrix.

Finally, the approximated execution state $\hat{I}_t$ is the weighted sum of the embedding vectors of all possible program input integers $c \in [-4, 4] \cap \mathbb{Z}$ (where $\mathbb{Z}$ is the set of all integers):

$$\hat{I}_{t,l} = \sum_{c \in [-4,4] \cap \mathbb{Z}} \mathbb{P}[I_{t,l} = c] E_{io}(c)$$

Here $E_{io}(c)$ denotes the embedding vector of the list value $c$. At the next program decoding step, $\hat{I}_t$ will be fed into the encoder to replace the previous input list $\hat{I}_{t-1}$.

### B.3.2 Implementation for Karel Domain

Similar to our restricted C domain, in our latent executor implementation for Karel domain, $\hat{I}_{t,l}$ is also the weighted sum of all possible execution states. As described in Appendix A, each Karel state describes the following variables: (1) $(\text{robot}_X, \text{robot}_Y)$ denotes the position of the Karel robot, where $0 \leq \text{robot}_X, \text{robot}_Y < 18$; (2) $\text{robot}_{dir} \leq \{\text{North, South, West, East}\}$ denotes the robot orientation at $(\text{robot}_X, \text{robot}_Y)$; and (3) the number of markers in each grid. Therefore, we train 3 predictors on top of $\text{LSTM}_E$ to predict these variables: (1) a trainable layer that outputs a $(18 \times 18)$-dimensional vector, representing the robot position; (2) a trainable layer that outputs a 4-dimensional vector, representing the robot orientation; and (3) an LSTM that generates an 11-dimensional vector at each step, representing the number of markers in each grid. We apply the softmax to all output vectors to obtain the probability distributions of different variables.

Afterward, we combine the outputs of the predictors to construct a $16 \times 18 \times 18$-dimensional vector representing the Karel state, according to Table 5, with the value of each dimension in $[0, 1]$. Note that Karel programs can not change the grid boundary and obstacles, thus we apply a mask on the predicted intermediate execution states to ensure that the features representing the grid boundary and obstacles remain the same, which are the last 2 dimensions described in Table 5.

Table 6: Results of iterative retraining on Karel dataset.

| Iters | 100% | 10% | 20% | 30% | 40% | 50% |
|-------|------|-----|-----|-----|-----|-----|
| | | Generalization Accuracy | | | | |
| 1 | 86.04% | 70.92% | 75.16% | 78.84% | 80.88% | 82.08% |
| 2 | 89.28% | 76.20% | 78.40% | 81.08% | 82.40% | 83.40% |
| 3 | 89.36% | 78.12% | 81.20% | 83.68% | 84.24% | 86.32% |
| | | Exact Match Accuracy | | | | |
| 1 | 39.40% | 36.20% | 37.20% | 38.36% | 40.20% | 40.04% |
| 2 | 41.56% | 37.24% | 37.28% | 39.24% | 39.72% | 39.16% |
| 3 | 41.16% | 36.56% | 38.16% | 38.68% | 38.72% | 39.64% |

Table 7: Results of iterative retraining on C dataset.

| Iters | 100% | 10% | 20% | 30% | 40% | 50% |
|-------|------|-----|-----|-----|-----|-----|
| 1 | 55.2% | 11.9% | 26.4% | 39.1% | 45.2% | 48.5% |
| 2 | 56.0% | 39.6% | 43.9% | 48.7% | 51.9% | 54.1% |
| 3 | 56.5% | 41.7% | 44.4% | 49.4% | 52.8% | 54.4% |

## C    Implementation Details

All encoders and decoders in our models are 2-layer bi-directional LSTMs with the hidden size of 512. The embedding size is 1024. We use the Adam optimizer [26] for training. The learning rate starts from 1e-3, and is decayed by 0.9 for every 6000 timesteps. The batch size is 8. The training converges in 200K batch updates. The norm for gradient clipping is 5.0. All models are trained on a single GPU. The beam size is 64 for evaluating the model performance, and is 8 for iterative retraining due to the large size of the training set.

About the implementation of the Property Signatures [39], we further illustrate the key difference between our adaption for the restricted C domain and the original implementation in [39] with the following example. Suppose an input-output pair is $([-4, 3, 1, 2, 1], [-4, 3, 3, 3, 3])$, when the feature is "Input == Output?", the corresponding property signature is "False" according to the implementation in [39], while the signature is "[True, True, False, False, False]" in our adapted implementation. Compared to the original implementation of property signatures, our adaptation better reveals which specific list elements are manipulated in the program. This modification makes our implementation of property signatures a much stronger baseline for the restricted C domain, because our C programs do not always perform the same manipulation steps over all elements in the input list, and sometimes change the values of only a subset of the input numbers.

## D    More Results of Iterative Retraining

Figure 10 presents more examples of predicted correct programs that are more concise than the randomly generated ground truth programs on C dataset.

Figure 11 presents more examples of predicted correct programs that are more concise than the randomly generated ground truth programs on Karel dataset. Note that the predicted Karel program is not semantically equivalent to the annotated ground truth in many cases. The main reason is because the randomly generated ground truth program might include redundant branching statements, i.e., the conditions always evaluate to true or false for all program inputs in the specification and the held-out test cases.

We present the numerical results of iterative retraining on Karel and C benchmarks in Table 6 and Table 7 respectively.

```
I1: [2, 4, 1, 2, -3]
O1: [2, 4, 3, 2, -3]
I2: [1, 0, 1, -3, 4]
O2: [1, 0, 3, -3, 4]
I3: [2, 2, -4, 2, 0]
O3: [2, 2, 3, 2, 0]
I4: [0, -2, 3, 1, 3]
O4: [0, -2, 3, 1, 3]
I5: [-2, 1, 4, 0, 0]
O5: [-2, 1, 3, 0, 0]
```

```c
int * func_1(int a[])
{
    int p_0 = 4;
    int l_7 = 2;
    int l_8 = 4;
    a[l_7] = 3;
    a[l_8] = a[p_0];
    return a;
}
```

```c
int * func_1(int a[])
{
    int p_0 = 2;
    a[p_0] = 3;
    return a;
}
```

```c
int * func_1(int a[])
{
    int p_0 = 2;
    int l_10 = 0;
    int l_1 = 4;
    l_10 = 2;
    for (p_0 = 2; p_0 >= 1; p_0--)
    {
        a[p_0] = 3;
        a[p_0] = 2;
        if (a[p_0])
            break;
        a[p_0] = a[l_1];
        a[p_0]++;
    }
    return a;
}
```

```
I1: [3, 1, 3, -2, -4]
O1: [3, 1, 2, -2, -4]
I2: [2, 0, -1, -1, 3]
O2: [2, 0, 2, -1, 3]
I3: [2, 0, -1, 4, 0]
O3: [2, 0, 2, 4, 0]
I4: [-2, -1, 3, 2, -4]
O4: [-2, -1, 2, 2, -4]
I5: [-4, 0, 3, 0, 1]
O5: [-4, 0, 2, 0, 1]
```

```c
// Training on random programs
int * func_1(int a[])
{
    int p_0 = 2;
    int l_7 = 2;
    a[l_7] = 2;
    return a;
}

// After iterative retraining
int * func_1(int a[])
{
    int p_0 = 2;
    a[p_0] = 2;
    return a;
}
```

```c
int * func_1(int a[])
{
    int p_0 = 0;
    int l_10 = 3;
    for (p_0 = 4; p_0 >= 0; p_0--)
    {
        a[p_0] = 3;
        a[p_0] = a[p_0];
        a[p_0] = 1;
        if (a[p_0])
            break;
    }
    a[l_10] = a[l_10];
    a[l_10] = a[p_0];
    return a;
}
```

```
I1: [0, 4, 0, 4, 2]
O1: [0, 4, 0, 1, 1]
I2: [4, 0, 1, 1, 4]
O2: [4, 0, 1, 1, 1]
I3: [3, 2, 3, 0, 0]
O3: [3, 2, 3, 1, 1]
I4: [1, 1, 4, 0, 4]
O4: [1, 1, 4, 1, 1]
I5: [1, 3, 0, 1, 1]
O5: [1, 3, 0, 1, 1]
```

```c
int * func_1(int a[])
{
    int p_0 = 4;
    for (p_0 = 3; p_0 <= 4; p_0++)
    {
        a[p_0] = 1;
    }
    return a;
}
```

```c
int * func_1(int a[])
{
    int p_0 = 0;
    int l_11 = 3;
    for (p_0 = 2; p_0 >= 1; p_0--)
    {
        for (int p_1 = 4; p_1 >= 3; p_1--)
        {
            a[p_1] = 4;
        }
    }
    a[p_0] = a[l_11];
    return a;
}
```

```
I1: [0, 3, -1, 0, 0]
O1: [4, 3, -1, 4, 4]
I2: [4, -3, 3, 4, 2]
O2: [4, -3, 3, 4, 4]
I3: [-4, 1, 0, 4, -2]
O3: [4, 1, 0, 4, 4]
I4: [0, 4, 3, 0, 4]
O4: [4, 4, 3, 4, 4]
I5: [2, 2, 0, 3, 2]
O5: [4, 2, 0, 4, 4]
```

```c
int * func_1(int a[])
{
    int p_0 = 3;
    int l_7 = 0;
    a[l_7] = 4;
    for (p_0 = 4; p_0 >= 3; p_0--)
    {
        a[p_0] = 4;
    }
    return a;
}
```

```c
int * func_1(int a[])
{
    int p_0 = 4;
    for (p_0 = 1; p_0 >= 0; p_0--)
    {
        a[p_0] = 1;
        for (int p_1 = 2; p_1 >= 1; p_1--)
        {
            a[p_1] = 4;
            a[p_1] = a[p_0];
            if (a[p_1])
                break;
        }
    }
    return a;
}
```

```
I1: [1, 0, 0, 4, -3]
O1: [1, 1, 1, 4, -3]
I2: [-3, 0, 0, -2, 4]
O2: [1, 1, 1, -2, 4]
I3: [0, 2, -2, 4, -3]
O3: [1, 1, 1, 4, -3]
I4: [4, -2, 0, -2, 0]
O4: [1, 1, 1, -2, 0]
I5: [0, 2, -4, 2, 2]
O5: [1, 1, 1, 2, 2]
```

```c
int * func_1(int a[])
{
    int p_0 = 1;
    for (p_0 = 2; p_0 >= 0; p_0--)
    {
        a[p_0] = 1;
    }
    return a;
}
```

Figure 10: More examples of predicted correct programs that are more concise than the randomly generated ground truth programs on C dataset. Left: input-output examples. Middle: the randomly generated ground truth program. Right: the predicted programs. Unless otherwise specified, the predicted programs come from the model trained on random programs.

```
def run():
  repeat (5):
    ifelse (rightIsClear):
      move
    else:
      move
    putMarker
```

```
def run():
  repeat (5):
    move
    putMarker
```

```
def run():
  move
  turnRight
  ifelse (noMarkersPresent):
    repeat (2):
      putMarker
  else:
    pickMarker
  repeat (5):
    turnRight
```

```
def run():
  move
  turnLeft
  turnLeft
  ifelse (markersPresent):
    pickMarker
  else:
    putMarker
    putMarker
```

```
def run():
  pickMarker
  move
  ifelse (not rightIsClear):
    putMarker
    move
  else:
    move
    putMarker
    while (not rightIsClear):
      move
      putMarker
  putMarker
  turnRight
  move
```

```
def run():
  pickMarker
  move
  move
  putMarker
  putMarker
  turnRight
  move
```

```
def run():
  move
  turnRight
  repeat (5):
    pickMarker
    putMarker
```

```
def run():
  move
  repeat (4):
    pickMarker
  turnRight
```

```
def run():
  move
  ifelse (markersPresent):
    ifelse (frontIsClear):
      putMarker
    else:
      pickMarker
  else:
    while (rightIsClear):
      turnRight
  repeat (2):
    repeat (2):
      putMarker
    turnLeft
```

```
def run():
  move
  while (leftIsClear):
    turnLeft
  repeat (4):
    putMarker
```

```
def run():
  putMarker
  move
  ifelse (not leftIsClear):
    putMarker
  else:
    turnRight
  if (rightIsClear):
    pickMarker
```

```
def run():
  putMarker
  move
  putMarker
  if (rightIsClear):
    pickMarker
```

```
def run():
  while (not rightIsClear):
    while (noMarkersPresent):
      putMarker
    move
    turnLeft
    ifelse (rightIsClear):
      while (noMarkersPresent):
        putMarker
        turnLeft
        turnLeft
        move
      turnLeft
      move
    else:
      turnLeft
  turnLeft
  turnLeft
  repeat (4):
    move
  turnLeft
```

```
def run():
  putMarker
  move
  if (noMarkersPresent):
    putMarker
    turnRight
    move
    putMarker
    turnRight
  while (not frontIsClear):
    turnRight
    move
    turnRight
  repeat (3):
    move
  turnLeft
```

```
def run():
  repeat (6):
    if (markersPresent):
      repeat (9):
        pickMarker
    putMarker
    move
  putMarker
  turnRight
```

```
def run():
  repeat (6):
    putMarker
    move
    putMarker
    turnRight
```

```
def run():
  turnRight
  move
  turnRight
  move
  while (not rightIsClear):
    move
  pickMarker
  ifelse (not leftIsClear):
    move
  else:
    move
```

```
def run():
  turnRight
  move
  turnRight
  move
  pickMarker
  move
```

```
def run():
  while (frontIsClear):
    ifelse (markersPresent):
      move
    else:
      putMarker
```

```
def run():
  while (frontIsClear):
    putMarker
    move
```

Figure 11: Examples of predicted correct programs that are more concise than the randomly generated ground truth programs on Karel dataset. 1st and 3rd columns: the randomly generated ground truth programs. 2nd and 4th: the corresponding predicted programs. The predictions come from the model trained on random programs.