# OpenReview forum: "Latent Execution for Neural Program Synthesis Beyond Domain-Specific Languages"
_NeurIPS.cc/2021/Conference — NeurIPS 2021 Poster_

### Official Review · Reviewer_KoBE · 2021-07-16

**Rating:** 6
**Confidence:** 4

**Summary:**

The goal of this paper is to perform program synthesis, taking inspiration from recent advances in execution-guided neural program synthesis in situations where execution of partial programs is not immediately possible.

As stated by the paper, the main contributions are
1) combining two types of representations - one which models the mapping from spec to program tokens, as in RobustFill (and other neural synthesis work), and the other which models the “hypothetical input signal for the remaining partial program to execute to get the desired output”

2) demonstrating that iterative retraining can lead to better modeling of code and improve sample efficiency.

3) showing that short C code can be synthesized from IO examples through the above two techniques.

**Ethical Concerns:**

None.

**Limitations And Societal Impact:**

See discussion of limitations above in main review, which I think do need to be more clearly addressed.

No negative societal impact.

**Main Review:**

Originality & significance:  The technique is original. This work addresses important questions in neural program synthesis; namely, how to adapt the successful execution-guided neural program synthesis to domains where intermediate execution is impossible or infeasible.
This work provides further relaxations from Nye et al. (2021), gaining from an approach inspired by execution-guided synthesis, but without learning neural functions for each DSL element.

Ablations are good, and seem to show that the partial executor is important for performance on the C dataset.

Iterative retraining seems to have a large effect at small dataset sizes; this effect diminishes at larger dataset sizes. For this reason, I do think the claim about iterative retraining would be stronger if it were evaluated on datasets build from real data, where extending the dataset automatically is not feasible. (See below — the dataset from Alet et al 2021 might be a good candidate). It seems that iterative retraining works because randomly generated programs are often more complicated than they need to be to solve the input-output examples (lines 226-230), but figure 3 and figure 7a seem to show that this mismatch can largely be solved by simply training on more synthetic data (which is always available for synthetic datasets). What would we see if we trained on real programs, where a) there isn’t always more data available but b) there may be less of a mismatch between the training data and the optimal programs?

Clarity:
- The paper is decently written, although there were some places where I was a little confused. In particular, I’m confused about how the latent execution setup works in the Karel domain. An explanation of this seems to be missing from both the main text and the supplement. In the C domain, $\hat{I}_{t,l}$ seems to be defined by a sum over embeddings of possible states. In the Karel domain, you don’t perform a sum over all possible Karel states, correct? Similarly, what is the Karel analog of the precomputed operation table?


- The hand-designed nature of the operation table of possible operations seems like a limitation. How would it scale to a larger set of values and operations? To me this seems to be the biggest limitation of the approach. The paper doesn’t contain much discussion of these limitations; I think adding a limitations section would greatly increase clarity
 For example, here are some other questions that it would be helpful to see answered:
	- I don’t have a clear sense of when this technique is easily applicable vs not applicable. Is the system as implemented only applicable to C programs with a type signature of List —> List, or could other type signature be easily supported? For example, how much engineering would be required to include programs of type (Int, List) —> List or List —> Int?
	- Is the approach computationally tractable when the set of possible intermediate states is too large to enumerate over? What happens when the range of integers is increased from [-4, 4] to [-256, 256] (or even [-2^32, 2^32])? Is that a limitation of this approach, or is there an easy fix? If it’s a limitation, I think that’s fine, but it would be helpful if it were clearly stated.

Overall, I do believe that this is an interesting contribution to neural program synthesis which extends the applicability of the promising line of neural execution-guided synthesis work. I felt like I learned something new reading this paper, although I don’t have a clear sense why this technique works. I would recommend acceptance, but I do think that improving the clarity about the Karel domain and a more clear discussion of limitations is important.


Comments/Questions:
Here are questions that are not necessary for my vote of acceptance, but things I think would be interesting to explore:

- Is there evidence that the latent execution representations $\hat{I}$ actually predict execution well? Can you accurately predict the actual intermediate state from $\hat{I}$?

- How do these models compare against transformer models pre-trained on language? I’ve seen that pre-trained transformers can achieve good performance when fine-tuned on synthesis tasks.

  - I’d be very interested to see the results of this system on the new C++ dataset from “A large-scale benchmark for few-shot program induction and synthesis” (Alet et al, ICML 2021). That would be an interesting test of this system on real-world code written by humans. As discussed above, I also think that it would be very useful to test iterative retraining here, to see if the approach has a benefit when the training dataset is derived from real code as opposed to synthetically generated.



**Time Spent Reviewing:**

5

---

> ### Author Response · Authors · 2021-08-10
> **Response and clarifications**
>
> Thanks for your encouraging comments! Please see the common response about the discussion on the applicability and limitations of the operation predictor, and comparison to pre-trained Transformer models. We address other comments below.
>
> ### Evaluation on the dataset from Alet et al.
>
> Thanks for pointing out this work! Though the paper is relevant, we note that: (1) the authors haven’t released the dataset yet; and (2) the full paper is only available around the ICML 2021 conference (mid-July), thus we didn’t cite this work in our initial draft before the NeurIPS deadline (end of May). We will discuss this work in our revision, and will evaluate LaSynth on their dataset once it is available.
>
> ### Latent execution setup for Karel
>
> Similar to the C domain, for Karel, $\hat{I}_{t,l}$ is also the weighted sum of all possible states. As described in Appendix B, each Karel state defines the following variables: (1) (robot_X, robot_Y) denotes the position of the Karel robot, where 0 <= robot_X, robot_Y < 18 (the maximum grid size); (2) robot_dir ∈ {North, South, West, East} denotes the robot orientation at (robot_X, robot_Y); and (3) the number of markers in each grid. Therefore, we train 3 predictors on top of $LSTM_E$ to predict these variables: (1) a trainable layer outputs an (18×18)-dimensional vector, representing the robot position; (2) a trainable layer outputs a 4-dimensional vector, representing the robot orientation; and (3) an LSTM generating an 11-dimensional vector at each step, representing the number of markers in each grid. We apply the softmax to all output vectors. Afterward, we combine the outputs of these predictors to construct a (16×18×18)-dimensional vector representing the Karel state, according to Table 4, with the value of each dimension in [0, 1]. Note that Karel programs can not change the grid boundary and obstacles, thus we apply a mask on the predicted intermediate states to ensure that the features representing the grid boundary and obstacles stay the same. We will describe more details in the revision.
>
> As discussed in the common response about the operation predictor, we agree that the operation predictor is designed for program synthesis problems involving numerical reasoning, and thus not included in the Karel model. We will revise the paper structure to make this point clearer.
>
> ### Generalizability to more type signatures
>
> To support more type signatures, we can generalize the model to take lists of various lengths as the input, and we can also consider an input integer as a list with one element. We will discuss this as future work in the revision.
>
> ### Accuracy on predicting the intermediate execution states
>
> On the Karel benchmark, when feeding in the ground truth programs, LaSynth accurately predicts the execution states for 59.08% program inputs. The ground truth intermediate states are not available for the C domain. For predicting the full program outputs, given the ground truth C programs, the accuracy is 65.8%. We will add more discussion in the revision.

---

> > ### Comment · Reviewer_KoBE · 2021-08-25
> > **Re: response**
> >
> > Thank you for the thorough response.
> >
> > One more question about the Karel $\hat{I}$ value: The equation at the bottom of page 14 in the appendix shows, for the C domain, a sum over the list c = [-4, -3, -2, -1, 0, -1, -2, -3, 4], correct? For Karel, would the analogous sum be over all possible values *for a single grid location*? In other words, is the sum done independently for each 16-dim vector representing a particular grid location?
> >
> > With respect to the dataset from Alet et al., I didn't mean to imply that evaluation on that dataset would affect acceptance to NeurIPS, as it is clearly too recent to expect a comparison. I was just "thinking out loud" that it would be cool/interesting to do so, once it's available.
> > Sorry if this was interpreted as something necessary for this conference -- I didn't mean to add stress to the reviewing process!

---

> > > ### Author Response · Authors · 2021-08-25
> > > **More clarifications on the latent execution modeling for Karel**
> > >
> > > Thanks for the follow-up! For Karel, the analogous sum is over all possible numbers of markers $m$ in each grid location, i.e., $m \in [0, 10]$. While the number of markers in each grid is independent from other grids, the 16-dim vector representations of grid locations are not entirely independent from others. As described in Table 4 on page 15, the first 4 dimensions of each grid represent the robot location and orientation, and thus should be non-zero only for a single grid in each Karel state (there is only one robot in the grid world). Therefore, besides an LSTM that predicts the number of markers in each grid, we include 2 separate layers for predicting the robot location and orientation. We fill in non-zero values for the predicted robot location, and set the values to 0 in the first 4 dimensions of the remaining grids. Afterward, we fill in the 10 dimensions representing the number of markers for each grid. Finally, the remaining 2 dimensions in the vector representation denote whether the grid belongs to the boundary or obstacle, and the values of these 2 dimensions stay the same as the initial input Karel states, because Karel programs can not change the grid boundary and obstacles. We will make these implementation details clearer in the revision.
> > >
> > > Thank you again for mentioning the dataset from Alet et al.! We agree that evaluating our approach on this dataset is an interesting next step, and we will work on it when the dataset becomes available.

---

### Official Review · Reviewer_BQk7 · 2021-07-16

**Rating:** 6
**Confidence:** 4

**Summary:**

This paper introduces LaSynth, a novel neural architecture that jointly learns to synthesize programs and perform a kind of targeted latent program execution. LaSynth is applied to synthesize Karel programs and list processing programs in the C programming language. The paper also finds that one to two iterations of dataset refinement and iterative retraining can improve the dataset quality and model performance. The LaSynth approach, with iterative retraining, is found to outperform baselines both in the Karel and C domains.

**Limitations And Societal Impact:**

The authors describe the main technical limitation of the work, the limited scope of the C language that it can handle. The authors do not discuss potential societal impact, indicating in the checklist they don't anticipate any such negative impact.

**Main Review:**

This work draws inspiration from and combines ideas from prior works using execution traces for program synthesis, and performing neural execution in latent space. It combines the core ideas of RobustFill for program generation with latent program execution to learn a latent program trace. This novel combination of ideas allows LaSynth to achieve strong results in two domains and outperform RobustFill and a Property Signature-based approach.

Beyond being a novel combination of ideas, the use of this kind of latent program execution from program synthesis is novel too. The latent states in the latent executor represent an abstraction both of (1) the result of the execution so far and (2) what the remainder of the program would need to be in order to achieve the target output.

The paper builds on a rich body of work on execution-guided program synthesis. Overall, it frames itself well within this literature. However, for the part of the paper on dataset refinement, this is lacking. I encourage the authors to put their dataset refinement contribution in the context of existing literature. Data augmentation is a rich field and has been used for program synthesis [1], and noisy student training [2] bears resemblance to your iterative approach.

The paper backs up its claims about the newly proposed approach LaSynth and about iterative dataset refinement through clear experimental evidence. It finds in the Karel domain that LaSynth (both with and without dataset refinement) outperforms the baselines considered, which do an adequate job of covering the space of modern approaches to the tasks.

How are the approaches made comparable? Are they given equal total compute or wall clock time, or are an equal number of samples drawn from each? How would an enumerative search approach compare?

The paper further finds that one (sometimes two) iterations of dataset refinement improve performance further. There is a rich body of existing work on data augmentation and self-training methods, which this paper does not mention. I suggest investigating his line of work and placing your dataset refinement contributions (which, to my knowledge, are novel for program synthesis) in the context of this body of existing work. See e.g. "noisy student training" [2].

The paper is overall clear in is explanation of its contributions, the method details, and the experiments. The paper states the latent executor hidden states represent the hypothetical input to the remaining program to produce the target output. This intuition bears elaboration. A concern I have is that the remaining partial program during latent execution could be the second half of a loop body, which in traditional code does not have a single input, nor can it be executed in isolation. Explaining the intuition of the hidden states more clearly would benefit the paper.

Program synthesis is an important problem lately of increasing interest to the NeurIPS community. Increasing the complexity of the languages in which synthesis is applicable is important, and the approach in this paper does just that. I do find the importance of the claim that this is the first time C code to be less significant than it likely sounds, however. Though technically a correct claim, this paper is looking at a narrow domain of programs, such that the complexity of working with C programs is comparable to working in a DSL for the domain. Particularly the significance is diminished (relative to the broader language of the claim) because of the restricted space of values considered, which includes only lists of the integers in $[-4, 4]$.

This limitation in the C domain manifests in the method with the following concern. The operation predictor uses a pre-computed table of all integer and subtraction operations. This is only possible due to the restricted domain of values, and prohibits the method as is from being used for even modestly broader domains.

Could you clarify what is meant by "we compute the property value per element in input-output pairs" at line 273?

---

Typo: "an associate memory structure" should be "associative"

---

[1] Guiding Program Synthesis by Learning to Generate Examples https://openreview.net/pdf?id=BJl07ySKvS

[2] Self-training with Noisy Student improves ImageNet classification https://arxiv.org/abs/1911.04252


**Time Spent Reviewing:**

8

---

> ### Author Response · Authors · 2021-08-10
> **Response and clarifications**
>
> Thanks for your encouraging comments! Please see the common response about the complexity of our C domain, and the applicability and limitations of the operation predictor. We will fix the typos in the revision, and we address other comments below.
>
> ### Discussion of existing works on data augmentation and self-training
>
> Thanks for pointing out these works! We will add discussion of these works and other data augmentation and self-training approaches in the revision. Specifically, [1] augments the set of input-output examples as the program specification, while our work fixes the input-output examples, but refines the program. Incorporating both lines of work to further improve the data quality is promising future work. [2] proposes a self-training scheme called noisy student training, which improves the image classification performance on ImageNet. How to adapt the noisy student training to the program synthesis domain is also an interesting direction for future work.
>
> ### Intuition of hidden states for syntactically incomplete partial programs
>
> When the partial code is not executable, the hidden state does not directly correspond to a single execution result. Instead, we interpret it as a distribution of possible subsequent execution states, as presented on Line 525. For example, when the remaining program is the second half of the loop body, the latent execution state could incorporate the hypothetical execution results after finishing the loop body in different ways. In fact, an important motivation of latent execution is to provide the model with the execution signal even when the partial program is not executable. We will make this point clearer in the revision.
>
> ### Baseline details
> To compare with different baselines, we sample an equal number of programs from each model. As described on Line 538, the beam size of our model is 64 for evaluation. We will make it clearer in the main body in the revision.
>
> An enumerative search is not applicable to both Karel and C domains. Note that the program length can be up to 256. Considering that the program search space grows exponentially to the program length, even for the Karel language with tens of tokens in the vocabulary, the search space is much larger than famous hard two-player games (e.g., Go game has a state complexity of 10^170). In this case, an enumerative search can hardly solve any problem.
>
> About the implementation of the property signatures baseline, take the input-output pair ([-4, 3, 1, 2, 1], [-4, 3, 3, 3, 3]) as an example, when the feature is “Input == Output?”, the property signature is “False” based on the implementation of the original paper, while the signature is [True, True, False, False, False] in our adapted implementation. Compared to the original property signature, our adaptation better reveals which specific list elements are manipulated in the program, making it a stronger baseline than the original implementation for our C domain. We will describe the baseline implementation in more detail in the revision.

---

> > ### Comment · Reviewer_BQk7 · 2021-08-24
> > **Thanks for your response and clarifications**
> >
> > Thanks for your response and clarifications. Your use of per element property values is clearer now. Thanks for your thoughts on the interpretation of the latent execution states.

---

### Official Review · Reviewer_WZjE · 2021-07-17

**Rating:** 7
**Confidence:** 5

**Summary:**

This paper presents a neural program synthesis system, called LaSynth, that aims to generate programs in non-domain specific languages (DSLs) unlike much of the prior work. In particular, this paper focuses on using LaSynth for the C programming language. A core reason prior work in program synthesis has been generally restricted to DSLs is, as the authors note (and I agree with), because of the linguistic complexity of general-purpose languages and the computational intractability that is often associated with synthesizing programs in such languages.

A core novelty of LaSynth is that it uses two forms of conjoined/parallel representations, one of which uses a hypothetical input to help determine the remaining partial program to execute to get the desired output. This somewhat resembles BUSTLE’s approach of breaking down components into small pieces and then assembling them for a larger program (ICLR ’21), but the actual system-level approach in LaSynth, in my mind, is notably different from BUSTLE and other prior works that I’m familiar (some of which the authors seem to have missed in the prior work – NetSyn (Mandal et al., MLSys ’21), REPL (Ellis et al., NeurIPS ’20), etc.), but we'll get to that later.


**Limitations And Societal Impact:**

None.

**Main Review:**

Overall, I’m positive about this paper, even though I think it has some weaknesses (most are simple things that can be addressed in revising the paper). What I like the most about this paper is the authors intentionally move away from DSLs and toy languages (like Karel) and instead targeting program synthesis for real-world programming languages, like C, which is the primary target language for this paper. By migrating to such real-world languages, many challenges emerge (e.g., syntax correctness, computational complexity, multiple correct synthesized programs for the same problem, etc.). This complexity has historically been a major restriction in program synthesis research. A core reason for this is that each additional language token can result in upwards of an order of magnitude increase in the computational search space for the program synthesizer, amongst many other core challenges.

The authors address this computational tractability problem by using partial programs and hypothesized intermediate state IO representations to help the synthesizer (the Latest Execution Trace (LaET)). In my opinion, the core novelty of the work is in the LaET. An additional novelty is in the construction of the synthesizer system design, which uses parallel representations and iterative retraining to help prune unnecessary instructions and reduce the synthesized programs to programs that “work” to programs that “a human might write” (see Figure 1’s code transformation from the top left to the top right).

A weakness in the paper is they only seem to compare their results to two other systems (outside of variants of their own): Property Signatures and RobustFill. Both systems are fairly relevant and new-ish. However, newer systems include REPL (NeurIPS ’20), BUSTLE (ICLR ’21), and NetSyn (MLSys ’21). I don’t think it’s a showstopper for that this paper doesn’t include these systems as perhaps there wasn’t enough time to get access to them. However, there should probably be at least some footnote somewhere explaining why these systems weren’t include (e.g., code not available, published after the submission, generally different approach, etc.). I hope the authors update the paper accordingly here.

Please fix this: there is a *plethora* of work that utilize execution traces for ML outside of that being used in Karel, which is a toy programming language and are the only three papers you cite for this space, all of which come from Dawn Song’s research group at Berkeley. Examples of others that use execution traces for machine programming, just off the top of my head: NetSyn (MLSys ’21), REPL (NeurIPS ’20), AutoPerf (NeurIPS ’19), Ithemal (ICML ‘19). When I saw that the only works cited for program traces was all came from the same group at Berkeley, I was rather disappointed (especially seeing that the authors seem to be well-versed in the literature). I strongly recommend you not singularly cite *one* researcher’s group in an area as broad as program traces for ML that being covered by dozens of research groups. By doing this – citing three papers (and only three) all from the same group all around the same (toy) language – it creates a perception that either you are from this group and you are self-citing or you are not informed about the state-of-the-art. Either case is poor research acumen, unbefitting of a tier-1 venue like NeurIPS, in my opinion.

Minor nit: the citations are not properly capitalized (e.g., “Deepcoder: Learning to write programs” -> “DeepCoder: Learning to Write Programs”, “Imagenet: A large-scale hierarchical image database” -> “ImageNet: a Large-Scale Hierarchical Image Database”, “Bustle: …” -> “BUSTLE: …”, etc.). Please fix these minor grammatical errors for subsequent revisions of the paper.


**Time Spent Reviewing:**

4

---

> ### Author Response · Authors · 2021-08-10
> **Response and clarifications**
>
> Thanks for acknowledging our work! We will fix the citations in the revision.
>
> About your comments on discussing prior works, we would like to clarify that we did not intend to ignore any related work, and we are sorry that you feel this way. Among the references you mentioned, we have already cited and discussed REPL and BUSTLE ([12, 23]) in several paragraphs of our paper. In the following, we explain why we did not empirically evaluate the works you mentioned, and we will add a more detailed discussion of these works in the revision.
>
> ### REPL and BUSTLE
>
> In the paper, we discussed that REPL and BUSTLE require a program interpreter to provide an execution state for partial programs at each step. They also do not support loops and conditionals, thus adapting them to our tasks requires a lot of work. In addition, their code repositories are not available.
>
> ### NetSyn
>
> This work proposes to learn a fitness function for genetic algorithms, and their neural network fitness function utilizes execution traces for full programs as part of the input. They evaluated all models on the DeepCoder task, and they developed an interpreter for the domain-specific language. Their code repository is not available, and we need to develop an interpreter to obtain execution traces for C domain, thus adapting NetSyn to Karel and C domains requires some nontrivial amount of work.
>
> ### Autoperf and Ithemal
>
> Autoperf proposes a learning approach for software performance regression testing, and Ithemal develops neural networks for throughput estimation. Both of these works do not study the program synthesis problem, thus are not directly applicable to our tasks.

---

### Official Review · Reviewer_iKX1 · 2021-07-18

**Rating:** 4
**Confidence:** 4

**Summary:**

This work improves neural program synthesis of imperative programs with limited control-flow (conditionals and loops)
from input-output examples. It extends the established pipelines for the Karel task – recurrent program decoder with
attention over partial output and example encodings – with modeling of a latent variable that represents a possible
execution state of the program at that location. The latent variable is supervised at the end to match the desired
output state. The authors test the approach on Karel (with mixed results), and on a randomly generated dataset of C
programs of similar complexity (where it substantially outperforms the baselines).

**Limitations And Societal Impact:**

No explicit limitations considered. The authors might consider discussing:
- what are the limits of "non-domain-specific" language constructs that latent execution might model successfully;
- the applicability of operation predictors;
- how might one generalize state embedding to settings where the state "type signature" is not uniform.

**Main Review:**


The key motivation of this work is to generalize execution-guided neural program synthesis to settings where step-wise
interpretation to retrieve actual program state is hard. Modeling expected program state as a latent variable,
predicting it with a recurrent model, and supervising to match the real state when known (in case of synthesis, the
desired output) is a promising and natural step towards that goal. However, the way the work presents its argument,
evaluates it, and compares against baselines falls short of its stated goal.

## Dataset
The key dataset that demonstrates the strength of LaSynth "beyond domain-specific languages" is a new dataset of
randomly-generated C programs. Its strengths include:
- A clever generation technique that leverages Csmith, a well-known random program generator for compiler fuzzing;
- Focus on challenging control flow, including for loops over integer lists, local variables, and arithmetic conditionals.

However, the sole fact that the generated programs are in C does not make it "unlike DSLs". The generated programs are
in a limited subset of C, de-facto unifying features of two existing DSLs: Karel and DeepCoder. They take loops and
conditionals from Karel, and variables, lists, and arithmetic from DeepCoder. As a result, the programs really do not
highlight the challenges by which the work is motivated. While C is compiled rather than interpreted, tracing partial
programs in this subset (to implement e.g. [6] or [22]) only requires one to close all outstanding blocks (to make the
program complete) and dump variable states at the location of interest.

Some challenging program synthesis datasets already satisfy the authors' requirements of (a) real programming language,
(b) input-output examples, (c) complex control flow. For instance:
- SPoC: https://sumith1896.github.io/spoc/
- APPS: https://github.com/hendrycks/apps

What is the reasonable upper bound of performance on the C dataset? For instance, how often is it the case that example
inputs have less that 100% coverage of the ground-truth program, and thus do not fully specify the intent?


## Baselines

The most natural baseline for this work would be [22], motivated by the similar goal of representing semantics of
partial programs as a latent variables. However, the authors do not provide any experimental evaluation, stating that
"per-line execution of partial C programs is infeasible" (L81). While generally true, this is definitely untrue for the
authors' dataset (e.g., via [CompCert Interpreter](https://compcert.org/man/manual004.html) or an ad-hoc one – see
above). As such, _conceptual_ comparison of two approaches is feasible. It is also warranted, given that LaSynth does
not compare nearly as favorably to explicit execution-guided synthesizers [28] and [6] on Karel.

Generalizing the point above, the fact that no baseline is shared between Karel and C evaluations makes it difficult to
appreciate the contribution of LaSynth. It clearly outperforms syntax-only synthesizers, but its comparison to the space
of execution-guided synthesizers (latent or explicit) is yet unclear.

What are the standard errors in Tables 2 and 3? How many runs were trained?

## Arithmetic modeling

The operation predictor is, in short, a lookup table for arithmetic operations that relies on the fact that only small
integers are used in the C dataset. Its inclusion is puzzling for many reasons:
- It is a domain-specific optimization that only works for small-number arithmetic, yet presented as part of the generic
  program architecture (Section 3.2). What is its instantiation for Karel?
- Table 1 claims that LaSynth has "no domain-specific features", yet operation predictor for arithmetic between -4 and 4
  that works because the dataset only contains these numbers is _most assuredly_ a domain-specific feature.
- It boosts accuracy on the C dataset by less than 2% – likely statistically significant but clearly orthogonal to the
  main contribution of the paper.

## Miscellaneous

More related work:
- Some research on task-oriented dialog requires the learned conversational system to model the latent "state" of the
  dialog, see e.g. [Min et al. 2020; Zhang et al. 2020]. (Usually the belief state is non-latent and labeled with some
  logical forms., thus more analogous to [6]).
- Predicting the next state after execution of the next instruction has also appeared in robotics [Paxton et al. 2019].

The "Iterative Retraining" trick is neat in the context of program synthesis. More broadly it can be likened to classic
_bootstrapping_ of ML models, where previous-iteration model is used to produce labels for an unsupervised dataset to
train the next-iteration model. Here the model is generative yet multiple "correct" labels for the same instance exist,
so this version of bootstrapping can re-label existing data instead. Worth exploring the connection in Related Work.
The fact that this bootstrapping brings to light that many Karel programs permit equivalent simpler implementations is a
very interesting discovery, worthy of preview in the Introduction.

Please clarify whether Equation 6 is computed in symbolic space or (as $\hat{I}_T$ might suggest) in representation
space and consequently, what exactly is the used loss.

$\mathit{EXEC}_t$ in Figure 2c is only defined in the Appendix.

L302: "we make the first attempt of synthesizing C code from input-output examples for list manipulation" is unfair. It
is obvious untrue for non-neural synthesizers, but even for neural synthesis it neglects e.g. SPoC [Kulal et al. 2019]
and very recently (for Python rather than C) Codex [Chen et al. 2021].


## References
1. Min, Qingkai, et al. "Dialogue State Induction Using Neural Latent Variable Models." IJCAI 2020.
2. Zhang, Yichi, et al. "A Probabilistic End-To-End Task-Oriented Dialog Model with Latent Belief States towards Semi-Supervised Learning." EMNLP 2020.
3. Paxton, Chris, et al. "Prospection: Interpretable plans from language by predicting the future." ICRA 2019.
4. Kulal, Sumith, et al. "SPoC: Search-based Pseudocode to Code." NeurIPS 2019.
5. Chen, Mark, et al. "Evaluating Large Language Models Trained on Code." arXiv preprint arXiv:2107.03374 (2021).


**Time Spent Reviewing:**

6

---

> ### Author Response · Authors · 2021-08-10
> **Response and clarifications**
>
> Thanks for your constructive comments! Please see the common response about the complexity of our C domain, and the applicability and limitations of the operation predictor. We address other comments below.
>
> ## Complexity of C domain
>
> Please refer to the section “Complexity of C domain” in the main response, where we address your comments about the comparison to domain-specific languages, differences from recent works on natural language to code generation and large-scale pre-trained language models, and the possibility of implementing an ad-hoc interpreter for our restricted C domain.
>
> We want to further emphasize that although our work has not yet demonstrated good results for synthesizing full-fledged C code, we still make promising progress towards synthesizing code in a much more complicated language from input-output examples only. This is also acknowledged by other reviewers (e.g., WZjE).
>
> Program synthesis from input-output examples is a challenging problem that has been studied for decades. Taking FlashFill DSL [1] as an example, which won the Most Influential POPL Paper Award in 2020. Since its first release in 2011, people have spent a decade improving the synthesis performance in this language, including general-purpose model improvement and search heuristic tuning specialized for FlashFill. As shown in the table comparison in the common response, despite that the FlashFill DSL does not support language constructs such as loops and conditionals, and the language design is specialized for string processing, making progress in this DSL is still a long-term effort.
>
> While we acknowledge that our introduction should be toned down a bit with a clear specification of the restricted C domain, we did not claim at all that our work is the ultimate solution to fully solving the C code synthesis problem. We respectfully disagree that our work has to solve challenging competitive programming problems to prove its contributions, in particular when only input-output pairs are available. Instead, we make an initial attempt by proposing novel program synthesis techniques, which model the latent execution for partial programs, and significantly improve the performance over existing approaches. We consider improving our technique to support more complicated code syntax, achieve better performance and scale up with large models/datasets as important future work, and we hope that our work can encourage more people to join this effort.
>
> [1] Sumit Gulwani., Automating string processing in spreadsheets using input-output examples, POPL 2011.
>
> ### C interpreter development
>
> Please refer to the section “C interpreter development” in the main response. Despite that LaSynth doesn’t perform as well against execution-guided synthesizers [6] where a symbolic interpreter is available and gives all intermediate execution states for partial Karel programs, we argue that such detailed information can be expensive or infeasible to obtain for general programming languages, and LaSynth has broader applicability without such detailed information. In fact, it outperforms [28] when only the execution traces for full programs are used.
>
> ## Baseline comparison
>
> Because C and Karel tasks take different formats of input-output specification as the model input, utilizing the same model architecture for both domains is inapplicable. However, as discussed on Line 197-199, the high-level design of Bunel et al. (for Karel) and RobustFill (for C) is similar, except that Bunel et al. use a CNN for embedding Karel grid maps, while RobustFill uses an LSTM for embedding lists.
>
> Note that following prior works, we consider generalization as the main metric (rather than exact match accuracy), because the Karel benchmark only provides one single reference program for computing the exact match, which makes this metric heavily biased towards that reference.
>
> ## Generalizability to more type signatures
>
> When the type signature is not uniform, we can generalize the latent executor to produce a list with various lengths in an autoregressive way, where each element is the embedding vector of the execution output at a certain position. We will discuss this as future work in the revision.
>
> ## Other comments
>
> ### Upper bound performance
>
> For the C domain, we observe that the input-output examples already fully cover the ground truth programs, so the upper bound performance should be 100%. However, on Karel, we observe that the input-output examples might underspecify the ground truth program. Specifically, with iterative retraining, our Karel model passes the given input-output examples for 96.04% problems on the validation set. Compared to the accuracy when the program needs to pass both the given input-output examples and the held-out test cases, i.e., the generalization accuracy defined on Line 195-196, we observe that the given input-output examples do not fully cover the ground truth programs for at least 7% problems. We will add more discussion in the revision.
>
> ### Standard errors
>
> For each configuration with the best hyper-parameters, we train 5 models independently from different random initialization. In Table 2, the standard error is 0.22% for the exact match accuracy, and 0.12% for the generalization accuracy. In Table 3, the standard error is 0.1%. We will add these results in the revision.
>
> ### More discussion on related work
>
> Thank you for pointing out the connection to related work on task-oriented dialogues and robotics! We will add the discussion of these works in the revision. We are glad that you like our iterative retraining approach, and we will describe the connection to bootstrapping, and add more discussion of iterative retraining in the introduction.
>
> ### Equation (6) loss and $EXEC_t$
>
> As discussed on Line 524-525, $\hat{I}_T$ is the weighted sum of embedding vectors of all possible execution states when the program is complete. The loss is the cross-entropy loss for the logit predicting $P[\hat{I}_T=O]$, when executing on the provided program input and ground truth programs. Due to the page limit, we defer details of the latent executor (including the equation to compute $EXEC_t$) to Appendix A.2, and we will add more details in the main body in the revision.

---

> > ### Author Response · Authors · 2021-08-19
> > **Has our response addressed your concerns?**
> >
> > Hi Reviewer iKX1, we would be grateful if you can confirm whether our response has addressed your concerns, and let us know if any issues remain. In the following, we recap the key points of our response:
> >
> > 1. Different from recent works on synthesizing code from text descriptions using pre-trained large-scale Transformer-based language models (released after NeurIPS submission deadline), our work focuses on program synthesis from input-output examples only, which is a totally different specification and also a long-standing important challenge in the program synthesis domain.
> >
> > 2. The restricted C domain used in this paper is still much more complicated than existing domain-specific languages such as DeepCoder and Karel. Without using any human-designed codebase, we have made promising progress using only input-output examples.
> >
> > 3. In contrast to what the reviewer states, implementing an ad-hoc interpreter, even for our restricted C domain, is difficult and inefficient. First, the partial program (like “for (“) is not executable due to the syntax design of C. Furthermore, even if such an ad-hoc interpreter can be built, for a different program language, similar efforts need to be repeated, which is highly inefficient. On the other hand, LaSynth has broader applicability without requiring such detailed information.
> >
> > 4. We discuss the applicability and limitations of the operator predictor. Despite that our current implementation of the operation predictor is specialized for the restricted C domain, we make an initial attempt to improve the numerical reasoning ability of program synthesis models, and demonstrate its effectiveness. A general arithmetic operator is out of the scope of this paper, and is left for future work.
> >
> > We are looking forward to your feedback!

---

> > > ### Comment · Reviewer_iKX1 · 2021-08-20
> > > **Please make the claims match the implementation**
> > >
> > > Thanks for the response, and for the great elaboration on individual points, especially comparison to prior work and perspective on claims.
> > >
> > > At the high level, my issue with the submission is not the work itself, but the claims. I would not have any issues with the limited-C dataset or with the single-digit operation predictor, if they were not advertised upfront with much stronger claims than they ended up being.
> > >
> > > In its current presentation, LaSynth (1) has a nice and novel central idea, which it (2) implements differently for two different domains, (3) without a unifying comparison between them. A version of LaSynth improves upon baselines on Karel. A different instantiation of LaSynth improves upon different, non-overlapping baselines on the new limited-C dataset. The second instantiation has an extra set of techniques (e.g., operation predictor). There is no single experiment that shows "This core idea, and only it, outperforms a strong (conceptually similar) baseline on two qualitatively different domains." This claim is instead supported by independent evaluations on different domain-specific baselines + by ablations of domain-specific techniques on limited-C. (The closest conceptual pair of baselines are indeed Bunel et al. for Karel and RobustFill for C, which do not involve any modeling of intermediate state, and so by now are not _strong_ baselines.)
> > >
> > > My set of suggestions for building an ad-hoc interpreter for limited-C was one direction to unify this comparison. Namely, it would create an evaluation environment for limited-C that facilitates running Exec [6], Execution-Guided Synthesis [28], or Latent Semantics [22] on it. Obviously, even slightly more complex subset of C would make such an interpreter infeasible. But this work's limited-C scope presents an _opportunity_ to conduct an apples-to-apples evaluation between latent-execution-guided and explicit-execution-guided synthesis, which is the key intriguing research question of this work.
> > > Here's how it could play out:
> > > - If LaSynth outperforms a strong execution-guided synthesizer on limited-C, it would show that latent execution is a powerful new technique, and would alone be a strong reason to accept the paper. Its slight underperformance on Karel could be explained by Karel being around longer with all the time/effort the researchers poured into it.
> > > - If LaSynth slightly underperforms on limited-C (just like on Karel), it would show that latent execution nearly approaches explicit-execution learning, within competitive range but without traces in the training data. This would also be a significant contribution, worthy of publication despite not SOTA on either domain.
> > > - And if LaSynth significantly underperforms on limited-C, that would raise questions and require further investigation.
> > >
> > > Right now, I don't know which is these stories is the reality, and thus unsure how to evaluate the paper's claims.
> > > There may be other ways to set up an apples-to-apples comparison with both domains. This is just the most natural one, albeit laborious.
> > >
> > > > We respectfully disagree that our work has to solve challenging competitive programming problems to prove its contributions, in particular when only input-output pairs are available.
> > >
> > > Respectfully, this misrepresents my review – nowhere did I suggest that LaSynth must "solve challenging competitive programming to prove its contributions". If I made the impression of the need to involve NL, I apologize – it's clearly out of scope; my comment concerns a different point.
> > >
> > > The paper's motivation can be summarized as "(a) real programming language, (b) input-output examples, (c) complex control flow" [my review, direct quote]. To match the scope of that motivation, both SPoC and APPS (a) contain input-output examples in addition to NL, (b) contain nontrivial sections of the dataset with simpler, short programs. Thus, the suggestion is to use exactly the same example-only setup as in limited-C, but on the subset of SPoC programs of appropriate length – _ignoring_ natural language.
> > >
> > > > our C domain still makes a first attempt towards synthesizing C programs only from input-output examples
> > >
> > > Quoting my review again: _"L302: "we make the first attempt of synthesizing C code from input-output examples for list manipulation" is unfair. It is obvious untrue for non-neural synthesizers, but even for neural synthesis it neglects e.g. SPoC [Kulal et al. 2019] and very recently (for Python rather than C) Codex [Chen et al. 2021]"_
> > > L302 (unlike your response) does not contain the word "only", hence SPoC is an obvious neural counter-example. Nor does it contain the word "neural". Again, the concern is not with the substance of the work, but with fixing the claim to make it precise and thus factually correct.
> > >
> > >
> > > ---
> > >
> > > I hope that this discussion helps restructure the writing to make it easier for the reader to appreciate domain-agnostic and domain-specific claims, and avoid the impression of overpromising. Namely, please consider:
> > > 1. Move the discussion of operation prediction to a section specific to limited-C. (Alternatively, you could keep the same abstract formalism, and also instantiate it in Karel to model a different set of API calls – which would make it much more interesting.)
> > > 2. Modeling numerical reasoning is indeed a well-known problem with plenty of challenges, baselines, and its own evaluation techniques, as you rightly mention. If LaSynth's operation predictor does not attempt to reach that bar, the paper should properly position it w.r.t. limited-C design choices. Yes, the proposed design _could_ be extended to a more general approach for numerical reasoning (your subtoken-based proposal being one way), but this would warrant an entirely new contribution and evaluation.
> > > 3. Compare the features of different DSLs to properly define the scope of limited-C (e.g. using the nice table in your common response), and tone down its introduction and novelty claim appropriately.
> > >
> > > Finally, I would like to reiterate that I find the core idea of this work original, elegant, and convincing. I could also say "effective", but the empirical evidence only partially establishes that (see above). With a more appropriate presentation of claims and limitations, it would already be a good publication for NeurIPS.

---

> > > > ### Author Response · Authors · 2021-08-21
> > > > **We will tone down the claims and clarify the scope**
> > > >
> > > > Thanks for the feedback! As also promised in responses we posted earlier, we will make the following changes to tone down our claims and clarify the limitations of our work:
> > > >
> > > > 1. In the introduction, include the table that compares the restricted C domain to existing domain-specific languages, and add more detailed discussion of the scope of this work.
> > > >
> > > > 2. Clearly put a limitation section stating that there is no ready-to-use token-by-token interpreter to get ground-truth execution trace for our restricted C domain to compare against, and therefore, our experiment indirectly supports our claim.
> > > >
> > > > 3. Move the discussion of the operation predictor to a subsection and make it clear that this component is designed for the restricted C domain.
> > > >
> > > > 4. Clarify the limitations of current implementation of the operation predictor, and discuss its extension for general-purpose numerical reasoning as future work.
> > > >
> > > > We will reference both SPoC [Kulal et al. 2019] and very recently Codex [Chen et al. 2021] (for Python rather than C, released after NeurIPS submission deadline) and clarify our contribution in the next revision.
> > > >
> > > > As you mentioned, building an ad-hoc interpreter for our restricted C domain is laborious. It is highly non-trivial to modify the interpreter you suggested to fit our needs, because the interpreter only takes complete C programs as the input, while LaSynth constructs execution traces for partial programs (like “for (“). Therefore, we may not have time to finish a strong execution-guided baseline with a dedicated interpreter for restricted C. However, we will try to add the discussion in the revision.

---

### Author Response · Authors · 2021-08-10
**Common response (part I, main contributions and the complexity of C domain)**

We thank all reviewers for their constructive feedback! We are glad that all reviewers find our latent execution approach novel and effective, and appreciate the design and empirical findings of our iterative retraining scheme. We address the common concern below.

## Main contributions

As acknowledged by all reviewers, our main contribution is to propose the latent execution techniques, called LaSynth, for neural program synthesis, which learns the intermediate execution states in scenarios when partial programs are not executable or an interpreter is not available. We demonstrate that LaSynth outperforms baselines that do not utilize the partial program execution, especially on our C benchmark. Although our work hasn’t supported the full C language yet, we note that program synthesis from input-output examples has been a long-standing challenge. Even for synthesizing programs in domain-specific languages, it often takes years or decades to make progress. Our work is by no means the ultimate solution fully solving the C code synthesis problem, and we consider improving our technique to support more complicated code syntax and achieve better performance for program synthesis from input-output examples as important future work. Meanwhile, our design of the latent execution framework is not limited to our input-output based program synthesis, but can potentially be used for other program synthesis problems from input-output examples combining other specifications such as natural language descriptions. We defer more detailed discussion about the C domain in the section “Complexity of C domain”.

Our second contribution is the iterative retraining scheme, which (1) improves the quality of the programs compared to randomly generated ones; and (2) further improves the model performance. According to the reviews, we will add more discussion about the iterative retraining in the revision.

About the operation predictor, we agree that this component is designed for program synthesis problems involving numerical reasoning, thus we only include it for C domain but not Karel. Meanwhile, its current implementation is limited to a bounded range of input numbers, and we will make these limitations clearer in the revision. However, the design of the operation predictor could potentially be scaled up to support a wider input range, and we defer more detailed discussion to the section “Applicability and limitations of operation predictor”.

## Complexity of C domain
One common concern from reviewers iKX1 and BQk7 is that our C domain is restricted. We agree that our current C benchmark only covers a subset of C, and we will tone down our claim about the significance with respect to the problem complexity, and make the limitation of our approach clearer in the abstract and introduction. However, our C domain is still much more challenging than existing domain-specific languages studied in neural program synthesis from input-output examples. Meanwhile, our work demonstrates promising results towards synthesizing code in general-purpose languages from input-output examples only, thus our contributions are complementary to the recent progress of large-scale pre-trained language models for program synthesis from natural language descriptions. We present a detailed comparison below.

### Comparison to domain-specific languages

| | Control flow | Variables | arithmetic |No helper functions | |
|-| -----------------| ------------ | -------------| -------------------------- | |
|Restricted C (ours) | Y |Y | Y |Y | |
|DeepCoder | N | Y | Y | N | |
|FlashFill | N | N | N | N | |
|Karel| Y | N | N | N | |

We presented an overview of the comparison of our restricted C domain and related existing domain-specific languages in the table. Although our C domain does not fully cover the C syntax, we note that it is not a simple combination of existing domain-specific languages (DSLs), e.g., DeepCoder and Karel, as commented by Reviewer iKX1. In particular, our C domain does not provide any domain-specific helper functions to simplify the implementation. Specifically, to avoid the need of synthesizing loops and conditional statements, DeepCoder defines domain-specific functions for list processing, e.g., Map and Filter, so that the program synthesizer can simply call these functions without including their internal implementation. Similarly, Karel language does not require variable assignment and arithmetic calculation, and defines domain-specific functions for checking conditions. Instead, our C domain does not provide any domain-specific helper functions, so the synthesizer needs to write all necessary routines from scratch, which requires the model to learn much more complicated and diverse code syntax.

Meanwhile, combining the features of existing DSLs, e.g., control flows, variable assignments and arithmetics, is not a trivial extension itself. Adding more syntax rules exponentially increases the program search space, which imposes significant challenges for program synthesis models. This is acknowledged by reviewer WZjE, and is consistent with the observations in prior works [1] that providing more information on the set of API calls to use leads to much better performance.

### Comparison to program synthesis from natural language and large-scale language models

Reviewer iKX1 points out some recent related works on synthesizing programs in C or Python, including SPoC (C), APPS (Python) and Codex (Python). First, we note that APPS and Codex were released around or after the NeurIPS paper submission deadline, which makes it impossible to reference by the time of our submission. While we agree that the C programs in our benchmark are less complicated than those datasets, the program specifications in all these works are in natural languages on human-written programs for competitive programming problems, which provides more information than input-output examples themselves. In particular, in SpoC, the specification is human-written natural language pseudocode that annotates the target program line-by-line, e.g., “set min_i to j if A[min_i] > A[j]”, which makes it unnecessary for program synthesis models to understand the input-output examples for prediction, and it is sufficient to only use input-output examples for selecting predicted programs. Therefore, our C domain still makes a first attempt towards synthesizing C programs only from input-output examples.

APPS and Codex demonstrate that large-scale language models (e.g., GPT-3) pre-trained on text corpus and/or GitHub can achieve non-trivial solve rates on real-world competitive programming problems from natural language descriptions. Although these are great achievements, program synthesis from input-output examples is also an important problem itself, and training on synthetic programs with the input-output specification format has several advantages. First, natural language descriptions are expensive to collect; e.g., APPS only includes ~10K natural language program specifications. On the other hand, input-output examples are much easier to generate automatically. Second, human-written programs are often subject to copyright protection, while synthetic programs are not. Therefore, dataset collection for input-output program synthesis problems can be much more scalable and requires less manual investigation.

Despite the restricted complexity of C programs in our evaluation, our latent execution framework is orthogonal to the advancement of pre-training techniques, and can be applied on top of them. We consider combining our technique with large-scale language pre-training and improving the complexity of the program domain as future work.

### C interpreter development
Reviewer iKX1 comments that it is possible to develop an ad-hoc interpreter for our C domain. While we agree that with a lot of manual effort, it is possible to dump some variable states when the partial program is syntactically valid, due to the complexity of our C domain, implementing an interpreter requires much more manual work than domain-specific languages in prior work, as recognized by Reviewer KoBE. In particular, as noted by Reviewer BQk7, the intermediate execution state is not available when the partial program is not executable, and this happens more frequently for C due to its syntax design (e.g., partial C code like “for (i=“). In such cases when a symbolic interpreter is unable to provide any execution result, LaSynth still produces some hypothetical execution signal to guide the program decoding. Furthermore, the main goal of our work is to demonstrate that without investing huge effort into developing an interpreter, by learning the latent representation of the execution results, LaSynth improves over baselines that do not utilize intermediate execution states. Finally, LaSynth is preferable when frequently interacting with an interpreter introduces significant overhead.

[1] Murali et al., Neural Sketch Learning for Conditional Program Generation, ICLR 2018.

---

> ### Author Response · Authors · 2021-08-10
> **Common response (part II, applicability and limitations of operation predictor)**
>
> ## Applicability and limitations of operation predictor
>
> A common question from reviewers is about the scalability of the operation predictor. We agree that the operation table is only able to enumerate the arithmetic operations over a bounded constant set. One way of extending our operation predictor to support potentially unbounded numerical calculation is to combine it with the subword tokenizer, which has been used in language models such as BERT [2]. Suppose a subword for a number contains at most 3 digits, then the operation table only needs to cover the arithmetic operations with the range [-1000, 1000], which is computationally feasible.
>
> Meanwhile, we agree that the operation predictor is designed for program synthesis problems involving numerical reasoning, and we only include it for C domain but not Karel. We will make it clear in the next revision that the operation predictor is still in its preliminary stage and can be domain-specific. Numerical reasoning has been a challenging task for deep neural networks, and there have been several studies showing that even the state-of-the-art large-scale pre-trained language models (e.g., GPT-3) are not capable of performing tasks requiring math calculation, despite their impressive performance on some other tasks [3, 4, 5]. Despite that our current implementation of the operation predictor is specialized for the C domain, we make an initial attempt towards improving the numerical reasoning ability of program synthesis models, and demonstrate that arithmetic modeling improves the model performance. Designing general-purpose number representation for better mathematical reasoning is out of the scope of this work, and we consider it as an important future direction.
>
> In the revision, we will clarify the limitations of the operation predictor, and add more discussion of the possibility of extending the current implementation to support a wider range of numerical inputs.
>
> [2] Devlin et al., BERT: Pre-training of Deep Bidirectional Transformers for Language Understanding
> [3] Wallace et al., Do NLP Models Know Numbers? Probing Numeracy in Embeddings, EMNLP 2019.
> [4] Brown et al., Language Models are Few-Shot Learners, NeurIPS 2020.
> [5] Zhang et al., Do Language Embeddings Capture Scales? EMNLP 2020 Findings.

---

### Decision · Program_Chairs · 2021-09-27

**Decision:**

Accept (Poster)

**Comment:**

OK, the situation with this paper is a bit unusual, owing to the fact that NeurIPS doesn't allow revisions during the reviewing period.
Basically all of the reviewers agreed that the core idea is good, but many of them had serious concerns with scoping and clarity.
I am going to recommend acceptance here, but I want to really strongly urge the authors to address the issues brought up by the reviewers in the final version of the paper. I don't actually have any authority to compel that outcome, so I'm relying on trust here - please do make these changes!